



# Spatiotemporal variation in the specific surface area of surface snow measured along the traverse route from the coast to Dome Fuji, Antarctica

Ryo Inoue[1,2], Teruo Aoki[2], Shuji Fujita[1,2], Shun Tsutaki[1,2], Hideaki Motoyama[1,2], Fumio Nakazawa[1,2], and Kenji Kawamura[1,2,3]

[1]Graduate Institute for Advanced Studies, SOKENDAI, Department of Advanced Studies, Tokyo 190–8518, Japan
[2]National Institute of Polar Research, Tokyo 190–8518, Japan
[3]Japan Agency for Marine–Earth Science and Technology, Kanagawa 237–0061, Japan

*Correspondence to*: Ryo Inoue (inoue.ryo@nipr.ac.jp)

**Abstract.**

To better understand the surface properties of the Antarctic ice sheet, we measured the specific surface area (SSA) of surface snow during two round-trip traverses between a coastal base near Syowa Station, located 15 km inland from the nearest coast, and Dome Fuji, located 1066 km inland, in East Antarctica from November 2021 to January 2022. Using a handheld
integrating sphere snow grain sizer (HISSGraS), which directly measures snow surface without sampling, we collected 215 sets of SSA data, each set comprising measurements from 10 surfaces along a 20 m transect. The measured SSA shows no elevation or temperature dependence between 15 and 500 km from the coast (elevation: 615–3000 m), with a mean and standard deviation of $25 \pm 9$ $\mathrm{m^2\,kg^{-1}}$. Beyond this range, SSA increases toward the interior, reaching $45 \pm 11$ $\mathrm{m^2\,kg^{-1}}$ between 800 and 1066 km from the coast (3600–3800 m). SSA shows significant variability depending on surface morphologies and
short-term meteorological events. For example, (i) glazed surfaces formed by an accumulation hiatus in katabatic wind areas show low SSA ($19 \pm 4$ $\mathrm{m^2\,kg^{-1}}$), decreasing the mean SSA and increasing SSA variability. (ii) Freshly deposited snow shows high SSA (60–110 $\mathrm{m^2\,kg^{-1}}$), but the snow deposition is inhibited by snow drifting at wind speeds above 5 $\mathrm{m\,s^{-1}}$. Our analyses clarified that temperature-dependent snow metamorphism, snowfall frequency, and wind-driven inhibition of snow deposition play crucial roles in the spatial variation of surface snow SSA in the Antarctic inland. The extensive dataset will
enable the validation of satellite-derived and model-simulated SSA variations across Antarctica.



## 1 Introduction

The specific surface area (SSA) of snow is the area of the ice–pore interface per unit mass of snow (Legagneux et al., 2002). Assuming spherical snow grains, the SSA relates to the effective radius ($r_{\mathrm{eff}}$), the area-weighted mean radius of the snow grains,
as follows (e.g., Flanner and Zender, 2006):

$$\mathbf{SSA} = \frac{S}{M} = \frac{3}{r_{\mathbf{eff}} \cdot \rho_{\mathbf{ice}}}. \tag{1}$$

Here, $S$ is the surface area of snow grains, $M$ is the mass of snow grains, and $\rho_{\mathrm{ice}}$ is the density of pure ice, which is 917 kg m$^{-3}$ at 0°C.

The snow SSA characterizes the evolution of snow metamorphism (Schneebeli and Sokratov, 2004; Domine et al., 2007) and has many polar glaciological applications. For example, the SSA is a key determinant for surface albedo (Wiscombe and Warren, 1980; Aoki et al., 2003) and microwave emissions (Brucker et al., 2011; Picard et al., 2013) because $r_{\mathrm{eff}}$ represents the optical grain radius controlling the light scattering and absorption properties of snow (Grenfell and Warren, 1999; Aoki et al., 2000). The SSA also affects the chemical composition of snow by determining the extent of ice–air interface where gas
molecules are adsorbed and also controls photochemistry within snow by determining the depth to which solar radiation penetrates (Domine et al., 2008; Zatko et al., 2016). Additionally, the SSA is related to the densification rate in the whole firn column; for example, fine-grained firn with a high SSA tends to have more grain interconnections and resistance to deformation compared to coarse-grained firn (Freitag et al., 2004; Fujita et al., 2009, 2014, 2016).

The SSA of near-surface snow changes through various processes depending on environmental conditions in the atmosphere and on the ice sheet surfaces. Snowfall or surface hoar formation deposits small grains on the surface, resulting in a SSA ranging between 30 and 200 m$^2$ kg$^{-1}$ at the surface (Domine et al., 2007; Libois et al., 2015). The freshly deposited snow grains become rounded (i.e., SSA decreases) with time under isothermal conditions through vapor transport, which is driven by curvature differences among snow crystals to reduce the surface free energy (Colbeck, 1980). The rate of SSA decrease
depends on the temperature that controls the amount of saturated water vapor (Legagneux and Domine, 2005; Taillandier et al., 2007). When diurnal and seasonal variations in insolation produce a vertical temperature gradient in near-surface snow, efficient vertical transport of water vapor occurs through vapor sublimation from warmer grains and condensation on colder grains, facilitating the development of large depth hoar (i.e., SSA decreases) (e.g., Yosida, 1955; Colbeck, 1983; Pinzer et al., 2012). In addition, grain growth is accelerated in the presence of liquid water at 0°C (Colbeck, 1980).


Wind influences snow SSA in complex ways. It can decrease snow SSA by sublimating fine needles or branches of dendric crystals in freshly fallen snow (Cabanes et al., 2002) and by eroding deposited snow, thereby exposing aged snow with a lower SSA (e.g., Lenaerts et al., 2017). Conversely, the wind can increase the SSA through the sublimation and fragmentation of



drifting snow crystals into small grains (Domine et al., 2009). Strong winds further contribute to snow redistribution and
heterogeneous deposition (Kameda et al., 2008; Picard et al., 2019). This process leads to variations in the SSA of surface
snow on a small spatial scale, reflecting varying degrees of post-depositional metamorphism. For example, within a 100 m
transect at Kohnen Station, Antarctica, the SSA shows a standard deviation (SD) of 11 m$^2$ kg$^{-1}$ around an average of 41 m$^2$
kg$^{-1}$ (Carlsen et al., 2017).

Satellite remote sensing can effectively monitor the spatial and temporal variations of snow physical properties over ice sheets.
Algorithms for retrieving the SSA of near-surface snow using near-infrared (NIR) light or microwave data have been developed
and applied to Antarctica (e.g., Scambos et al., 2007; Jin et al., 2008; Brucker et al., 2010) and Greenland (e.g., Hori et al.,
2007; Lyapustin et al., 2009). However, these algorithms typically assume the radiation interacts with a flat surface, which
introduces errors in the SSA retrievals. This is particularly true in Antarctica, where the presence of sastrugi – rough surfaces
with 0.1–0.5 m high undulations parallel to a predominant wind direction – complicates measurements (e.g., Warren et al.,
1998; Kuchiki et al., 2011). For example, satellite-derived SSA showed unrealistic diurnal variations because of the changing
relative angle between sastrugi and sunlight, leading to SSA between 5 and 330 m$^2$ kg$^{-1}$ at the South Pole (Kuchiki et al.,
2011). This underscores the need for ground-truth SSA data to improve satellite retrievals.

Several optical techniques can measure snow SSA, such as an IceCube (Gallet et al., 2009), an alpine snow specific surface
area profiler (ASSSAP) (Libois et al., 2015), and albedometers (e.g., Arioli et al., 2023). They measure the NIR reflectance of
snow and use a theoretical relationship between the reflectance and SSA to determine the SSA values (Wiscombe and Warren,
1980). In addition, X-ray computed tomography has been employed for analyzing the high-resolution 3-D microstructures of
snow, from which SSA is calculated (e.g., Schneebeli and Sokratov, 2004). With these techniques, the SSA of near-surface
snow has been measured at multiple sites in Antarctica, such as Kohnen Station (Linow et al., 2012; Proksch et al., 2015;
Carlsen et al., 2017), Dome Fuji (Inoue et al., 2024), the inland plateau of Wilkes Land (Calonne et al., 2017; Picard et al.,
2022) including Dome C (e.g., Brucker et al., 2011; Picard et al., 2014; Libois et al., 2014, 2015), and Adélie Land (Gallet et
al., 2011; Picard et al., 2022; Arioli et al., 2023) (Fig. 1a; see Table S1 for details of the studies).

Some studies have examined the environmental factors controlling the SSA of near-surface snow. For example, Libois et al.
(2015) measured the SSA at Dome C during two summers using a spectral albedometer and ASSSAP. They observed a
decrease in SSA from approximately 80 m$^2$ kg$^{-1}$ in late October to 30 m$^2$ kg$^{-1}$ in late January, a period during which the snow
temperature increases as the snowpack absorbs solar radiation. They also observed a constant high snow SSA (around 60 m$^2$
kg$^{-1}$) for several days due to a continuous surface hoar formation and a rapid SSA decrease during strong wind events, which
might erode the surface and expose aged snow. Arioli et al. (2023) assessed the wind effect in more detail through combined
observations of near-surface snow SSA, surface height, and snow transport by drifting at two windy locations in Adélie Coast
from 2017 to 2021. They found that winds inhibit snow deposition for half of the observed snowfall events, while wind-driven



snow drifting causes a concomitant deposition of fine grains into near-surface snow, compensating for a SSA decrease due to snow metamorphism. Gallet et al. (2011) investigated the spatial variation of near-surface snow SSA between Dumont

D'Urville and Dome C through SSA measurements on 21 pit walls using IceCube. They found that the SSA in the top 0.1 m is higher (30–40 m$^2$ kg$^{-1}$) between 600 km from the coast and Dome C than the region between 0 and 600 km from the coast (approximately 20 m$^2$ kg$^{-1}$). They also suggested that the SSA at Dome C, which is higher than expected from an empirical SSA–density relationship for seasonal snow, is attributable to the long-term wind-driven fragmentation and sublimation of snow grains without burial (Domine et al., 2009).


However, most previous studies have focused on a few sites or only observed one spot at each site using a pit wall or a firn core (Fig. 1a and Table S1). Consequently, essential data and understanding of the wide-area distribution of snow SSA are still lacking. For example, (i) SSA measurements using pit walls or core samples often lack the data in the top few centimeters, a depth which significantly impacts surface albedo (e.g., Wiscombe and Warren, 1980; Aoki et al., 2003). (ii) Considering the

small-scale variability of near-surface snow SSA within a horizontal extent of tens of meters (Libois et al., 2014, 2015; Carlsen et al., 2017), a single-spot measurement is not sufficient to provide representative SSA at a site, especially when compared to satellite retrievals whose spatial resolutions range from 150 to 500 m (Scambos et al., 2007; Jin et al., 2008). (iii) It remains uncertain which processes – snowfall, surface hoar formation, temperature-dependent metamorphism, and wind-driven snow erosion or fragmentation, observed at specific sites in previous studies – are crucial in controlling the spatial and temporal

variations of near-surface snow SSA across Antarctica.

One reason for the sparse SSA data for surface snow might be the time required for measurements, which involves careful sampling procedures (Gallet et al., 2009) or the setting up of instruments on the surface (Libois et al., 2015). Recently, to increase the efficiency of SSA measurements in the field, a handheld integrating sphere snow grain sizer (HISSGraS) has been

developed (Aoki et al., 2023). HISSGraS enables quick SSA measurements by directly measuring snow surfaces without the need for sampling or setting up any instruments on the surface. It also eliminates the need to adjust the temperature-sensitive laser light source to the ambient temperature before measurements, which typically requires about 30 minutes (Aoki et al., 2023).

This study aims to increase the accessibility of spatially representative SSA data for surface snow and reveal its wide-area distribution in Antarctica. To achieve this goal, we measured the SSA of surface snow using HISSGraS during two round-trip traverses between the coast near Syowa Station and Dome Fuji in the summer of 2021–2022. We provide the first detailed view of the wide-area distribution in the SSA of surface snow in Antarctica based on ground-based observations, with an extensive dataset from ~ 2150 spots. Using the new data, we discuss the environmental factors and processes that primarily

control the spatial and temporal variations in the SSA of surface snow.





## 2 Methods and data

### 2.1 Study area

In East Antarctica, snowfalls occur due to the activity of offshore cyclones or their blockage by high-pressure ridges, which
transport warm, moist air toward the continent (e.g., Souverijns et al., 2018; Turner et al., 2019). The snowfall gradually decreases toward the interior plateau due to the orographic lifting of the moist air (e.g., Palerme et al., 2014). The deposited snow can be redistributed by snow drifting, particularly toward the coast, over scales of hundreds of kilometers in katabatic wind areas (e.g., Lenaerts and van den Broeke, 2012). The interaction between snowfall and wind-driven redistribution influences the spatial variation of the surface mass balance (SMB) and surface morphologies (Watanabe, 1978; Furukawa et
al., 1996; Filhol and Sturm, 2015).

East Dronning Maud Land (DML) can be divided into three regions based on the surface morphologies: the coastal region (elevation: 500–2000 m), the katabatic wind region (2000–3600 m), and the inland plateau region (3600–3800 m) (Watanabe, 1978; Furukawa et al., 1996) (Fig. 1b). In the coastal region, frequent accumulation occurs due to offshore cyclones and snow
drifting from the interior by katabatic winds, resulting in high SMBs of up to 300 mm w.e. yr$^{-1}$ (Watanabe, 1978; Takahashi et al., 1994). The high accumulation leads to relatively flat surfaces with undulations of less than 0.3 m (Furukawa et al., 1996). In the katabatic wind region, the spatial pattern of snow deposition is primarily controlled by snow redistribution, which in turn depends on surface slopes that fluctuate between 0.1 and 0.5 degrees at intervals of 5–20 km. On relatively gentle slopes, snow is deposited as dunes and then exposed to continuous katabatic winds, resulting in sastrugi by erosion (Furukawa et al.,
1996). In contrast, on relatively steep slopes, katabatic winds accelerate and inhibit snow deposition, resulting in accumulation hiatuses and the formation of glazed surfaces (Furukawa et al., 1996). This selective accumulation in this region leads to significant variations in SMB, ranging between 0 and 200 mm w.e. yr$^{-1}$ (Takahashi et al., 1994). In the inland plateau region, snowfall or diamond dust is deposited under calm wind conditions, forming dunes without being eroded (Furukawa et al., 1996). The SMB decreases from 50 to 25 mm w.e. yr$^{-1}$ toward the dome area (Takahashi et al., 1994).




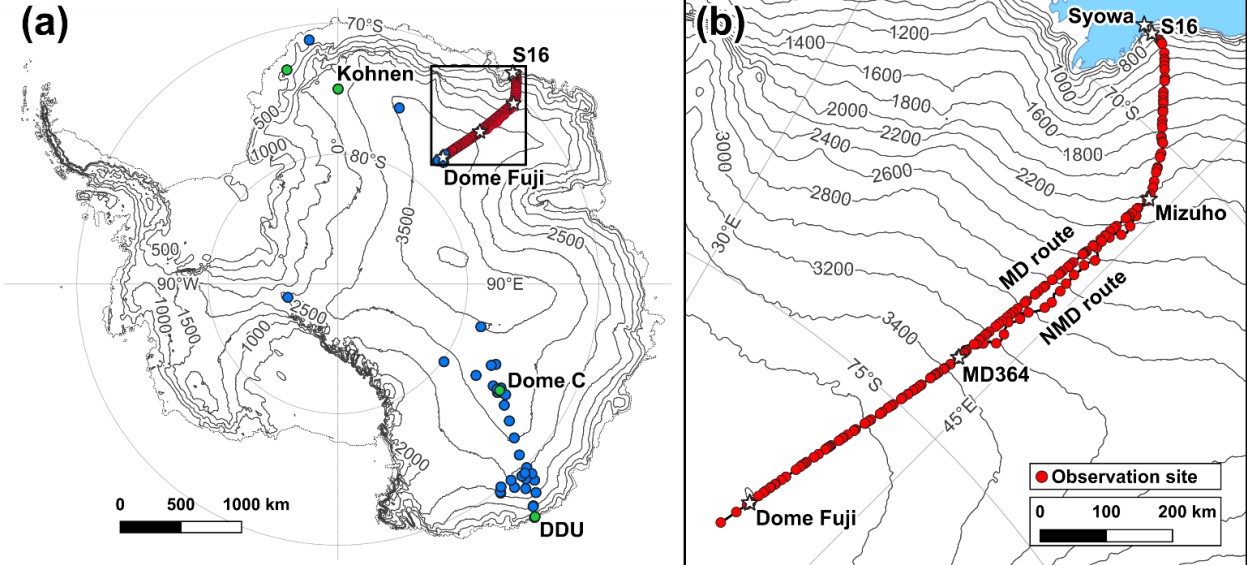

**Figure 1: Map of observation sites for snow SSA in Antarctica. (a) Topographic map of Antarctica. Contours indicate elevation (m) based on the CryoSat-2-derived elevation model referenced to WGS84 (Helm et al., 2014). Blue and green markers indicate observation sites for a vertical profile of snow SSA and the temporal variation of surface snow SSA, respectively (Table S1). Red markers indicate observation sites for surface snow SSA (10 different surfaces at each site) (this study). Stars indicate major sites along the traverse route mentioned in the text. (b) Enlarged view of the area enclosed by the rectangle in (a).**

### 2.2 Field observations

The field observations were conducted along traverse routes between the S16 site located 15 km from the nearest coast at an elevation of 615 m and the NDF site located 1066 km inland along the route (54 km south of Dome Fuji Station) at 3800 m (Fig. 1b). We conducted two round-trip traverses, totaling four traverses on the same path, from 12 November 2021 to 31 January 2022 (mainly for logistical reasons). We departed S16 on 12 November and arrived at Dome Fuji Station on 1 December. After traversing further to NDF on 3 December, we returned to S16 on 16 December. Following a 4-day stay at

S16, we started another traverse to Dome Fuji from 21 December to 5 January. Then, after a 13-day stay at Dome Fuji Station until 17 January, we returned to S16 on 31 January. During the last traverse, two observers individually measured the SSA between 932 and 278 km from the coast, partially along different routes: the MD route used in the previous three traverses and the NMD route, which was designed to avoid intense sastrugi areas (Fig. 1b).

We conducted observation activities for snow SSA at approximately 20 km intervals during each traverse. Additionally, we performed the activity twice daily (around 8:00 and 20:00 LT) at a fixed location near Dome Fuji Station from 5 to 17 January



to track the temporal variation of SSA. In total, we carried out 215 observation activities during the four traverses. The items measured during each activity are described below.

### 2.2.1 SSA measurement

We used HISSGraS for SSA measurements, which shares the same measurement principle as IceCube (Gallet et al., 2009) but offers advantages such as being lightweight, handheld, and capable of directly measuring snow surfaces without the need for sampling (Aoki et al., 2023). It employs an integrating sphere, the circular part of which (25 mm diameter) is a glass window. Inside, a laser diode and an InGaAs photodiode are attached. The laser diode emits NIR light at 1310 nm through the glass window, which is in direct contact with the snow surface. The penetration depth – the depth at which the light intensity reduces to $e^{-1}$ of its incident value – for NIR light at 1310 nm is approximately 8 mm for fresh snow and 9 mm for depth hoar with a SSA of 40 and 12 $m^2\,kg^{-1}$ and a density of 120 and 230 $kg\,m^{-3}$, respectively (Gallet et al., 2011). Therefore, HISSGraS provides a weighted average of the snow SSA over approximately the top 10 mm of near-surface snow (referred to hereafter as "surface snow SSA"). The InGaAs photodiode collects the light reflected by the snow surface. The measured light intensity is then converted to reflectance ($R$) using a calibration curve derived from the measurements on six reflectance standards (5–99 %). Since the calibration curve varies with ambient temperature due to the temperature sensitivity of the laser diode emission ($-1$ % $K^{-1}$), HISSGraS records the temperature close to the laser diode for every light intensity measurement, enabling the correction for the temperature dependence of calibration curves. Following Aoki et al. (2023), we constructed a calibration formula applicable to the temperature range observed during our study ($-35$ to $5°C$) (see Supplementary Note S1 and Fig. S1 for details). Finally, the calibrated $R$ is converted to SSA using a theoretical $R$–SSA relationship derived from a radiative transfer model that assumes spherical snow grains and employs Mie theory (Aoki et al., 1999).

At each observation site, we measured surface snow SSA at 10 different surfaces, spaced 2 m apart along a transect perpendicular to the predominant wind direction. For each of the 10 surfaces, we conducted five measurements by shifting the measurement positions by approximately 0.1 m and calculated their mean value. Data affected by accidental sunlight intrusion into the integrating sphere were excluded from the averaging. Such incidents were identified by irregularly high values of dark current, which were automatically measured with no laser illumination for all SSA measurements. Pressing the glass window of HISSGraS onto the surface did not leave deep traces. We carefully pressed the surface with the glass window for freshly deposited snow to fill the voids between snow grains, as Gallet et al. (2011) did, leaving traces a few millimeters deep.

The relative SD of the five measurements at a surface, which represents the random error in a SSA measurement, is 3.5 % (the average for ~ 2150 surfaces). The absolute error in the HISSGraS measurements is less than 23.0 %, which has been evaluated by comparing the HISSGraS data with the accurate SSA from the $CH_4$ adsorption method (Aoki et al., 2023).





### 2.2.2 Classification of surface morphologies

We classified the morphologies of all measured surfaces. Surface morphologies are broadly categorized into three forms from the viewpoint of SMB: depositional form, erosional form, and accumulation-hiatus form (e.g., Watanabe, 1978; Goodwin, 1990). Considering the time elapsed after snow deposition, which may relate to surface snow SSA, we further classified the surface morphologies into five types:


a) Fresh deposition surface (Fig. 2a). This includes precipitation particles freshly deposited homogeneously, in dunes, or in snowdrifts (Watanabe, 1978; Filhol and Sturm, 2015). Surfaces covered by snow that can easily be redistributed by wind due to their fragility were classified into this type.


b) Aged deposition surface (Fig. 2b). A depositional form that includes dunes and snowdrifts (Watanabe, 1978; Filhol and Sturm, 2015). Surfaces less likely to be redistributed due to their aged and hardened snow were classified into this type.

c) Erosion surface (Fig. 2c). An erosional form, resulting from wind-driven erosion and pitting of aged deposition surfaces (Watanabe, 1978; Goodwin, 1990). This form is distinguished from sastrugi by having a relatively flat surface with undulations less than approximately 0.1 m.


d) Sastrugi (Fig. 2d). An erosional form resulting from wind-driven erosion of large dunes, which leaves the hard part of dunes uneroded and exposed to strong winds for a long period. Surface undulations typically exceed 0.1 m (Goodwin, 1990; Furukawa et al., 1996; Filhol and Sturm, 2015).


e) Glazed surface (Fig. 2e). A long-term accumulation-hiatus form consisting of multi-layered crusts several millimeters thick (e.g., Watanabe, 1978). During summer, crust layers develop by the condensation of water vapor transported from subsurface depth hoar layers (Fujii and Kusunoki, 1982).





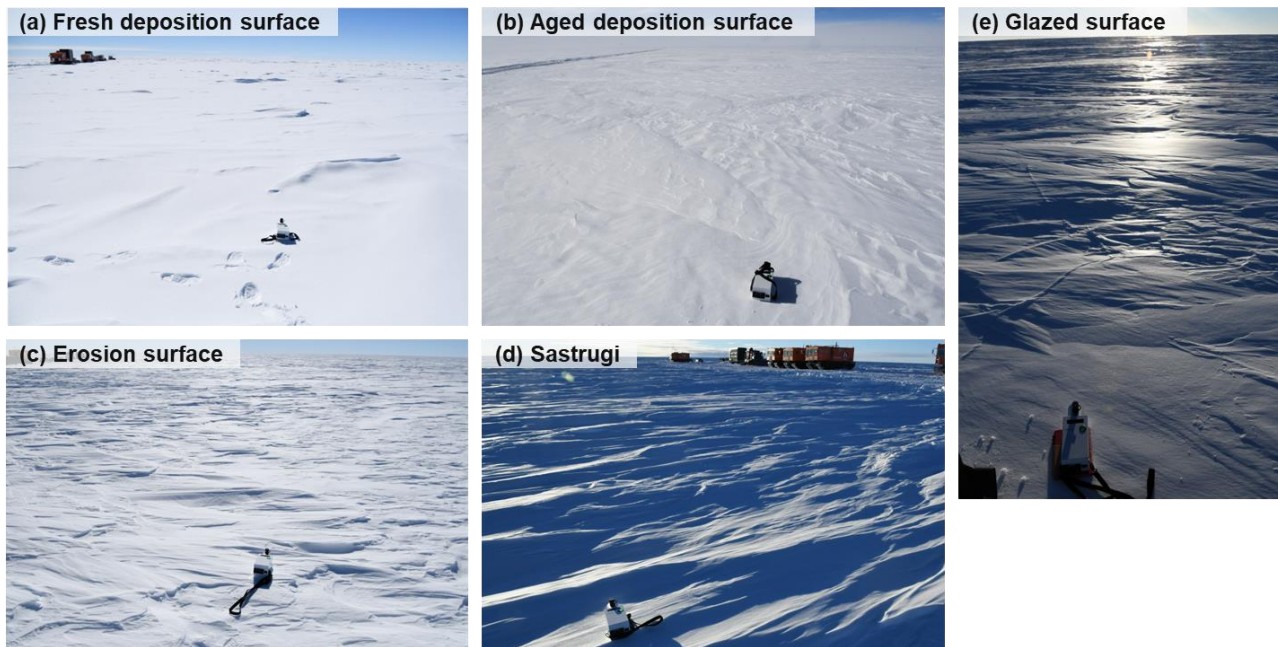


**Figure 2: Photographs of surface morphologies with HISSGraS (0.3 × 0.1 × 0.1 m³ volume) on the surface. (a) Fresh deposition surface (homogeneously deposited precipitation particles) at 584 km from the coast on 29 December 2021, (b) aged deposition surface (snowdrifts) at 987 km on 4 January 2022, (c) erosion surface at 225 km on 28 January 2022, (d) sastrugi at 382 km on 26 December 2021, and (e) glazed surface at 646 km on 26 November 2021.**


### 2.2.3 Weather observations

During each observation activity, we visually observed cloud cover, the presence of snowfall, and the presence of drifting snow. These parameters were also monitored in conjunction with air temperature using a handheld thermometer (TR-52S, T&D, Japan; accuracy of ±0.3°C) and mean wind speed using a handheld anemometer (Kestrel-5500, Mistral Instruments, 235 Japan; accuracy of ±3 %) three times daily (6:00–7:00, 12:00–13:30, and 19:00–20:30 LT) throughout the four traverses.

### 2.3 Automatic weather station data

To investigate the relationship between surface snow SSA and meteorological conditions, we utilized air temperature, wind speed, and air pressure data recorded at eight automatic weather stations (AWSs) installed along the traverse route. The stations 240 are S17 (16 km from the coast along the traverse route), H128 (94 km), Mizuho (278 km), MD78 (360 km), MD364a, MD364b (646 km), Dome Fuji (1024 km), and NDF (1066 km) AWS. S17 AWS is operated by the Japan Meteorological Agency (unpublished data). H128, MD78, MD364a, and NDF AWS are operated by the National Institute of Polar Research (NIPR)



(https://ads.nipr.ac.jp/real-time-monitors/, last access: 20 February 2024). Mizuho, MD364b, and Dome Fuji AWS are operated by the Antarctic Meteorological Research and Data Center, University of Wisconsin (UW) (https://amrdcdata.ssec.wisc.edu/, last access: 20 February 2024).

## 3 Results

### 3.1 Surface snow SSA between the coast and Dome Fuji

We describe the spatiotemporal variation of surface snow SSA during the four traverses between S16 and Dome Fuji and their relationships to the local weather conditions.

#### 3.1.1 First traverse

Surface snow SSA at 10 different surfaces of each observation site ranges from 10 to 85 $m^2 kg^{-1}$ during the first traverse, with higher values observed toward the interior (Fig. 3a and 3b; the right axis in Fig. 3b shows $r_{eff}$ inversely proportional to SSA for interested readers). In the coastal and lower katabatic wind regions, the surface snow SSAs are $29 \pm 15$ and $26 \pm 11$ $m^2 kg^{-1}$ (mean and SD), respectively (Table 1), showing no significant increase toward the interior (the slope of the linear regression for the 10-surface mean SSA is $-0.003 \pm 0.002$ $m^2 kg^{-1} km^{-1}$). Beyond these regions, SSA significantly increases to $34 \pm 9$ $m^2 kg^{-1}$ in the upper katabatic wind region and $46 \pm 8$ $m^2 kg^{-1}$ in the inland plateau region ($0.053 \pm 0.001$ $m^2 kg^{-1} km^{-1}$).

Surface snow SSA depends on surface morphologies (Fig. 3b). The fresh deposition surface, observed at 94 km from the coast, shows the highest SSA (70–85 $m^2 kg^{-1}$) among the five surface morphologies. The SSA of the erosion surface ($32 \pm 11$ $m^2 kg^{-1}$), primarily observed in the coastal region, is lower than that of aged deposition surface ($42 \pm 10$ $m^2 kg^{-1}$), primarily observed in the inland plateau region. The SSA of sastrugi ($29 \pm 8$ $m^2 kg^{-1}$), observed in the katabatic wind region, is similar to erosion surface. The glazed surface, observed in the katabatic wind region, shows the lowest SSA ($19 \pm 5$ $m^2 kg^{-1}$) among the five surface morphologies, resulting in higher SSA variability in the region (SD of 9–11 $m^2 kg^{-1}$) than in the coastal region (6 $m^2 kg^{-1}$, excluding fresh deposition surfaces) and the inland plateau region (8 $m^2 kg^{-1}$) (Table 1).

Air temperature at 12:00–13:30 LT decreases from −8°C to −29°C toward the interior, while wind speed measured at the same time ranges from 2 to 12 m s⁻¹, with higher values in the katabatic wind region compared to the coastal and inland plateau regions (Fig. 3c). Similar distributions for air temperature and wind speed are observed at 6:00–7:00 and 19:00–20:30 LT, with temperature biases of −6 and −2 °C, respectively (not shown).



Distinct variations in surface snow SSA were observed after two snowfall events at different wind speeds. First, a light snowfall
occurred at 94 km from the coast on 14 November with wind speed of ~ 6 m s$^{-1}$ (Fig. 3c). During this event, precipitation
particles were heterogeneously deposited on 30 % of the observed surfaces, increasing surface snow SSA to 70–85 m$^2$ kg$^{-1}$
(Fig. 3b). Second, a severe blizzard occurred at Mizuho on 17–19 November (marked by "A" in Fig. 3b), associated with a
blocking-high activity in Princess Elizabeth Land (see Fig. S2 for meteorological fields). At Mizuho AWS, a maximum wind
speed exceeded 20 m s$^{-1}$, and air temperature and pressure increased during the blizzard event (Fig. 4b; see Fig. S3 for the data
of the eight AWSs). After this event, no fresh deposition surfaces of precipitation particles were observed. Instead, 50 % of
the observed surfaces turned into glazed surfaces with low SSA (Fig. 4b). Consequently, the 10-surface mean SSA decreased
from 31 to 21 m$^2$ kg$^{-1}$. The low mean SSA around 20 m$^2$ kg$^{-1}$ was also observed between 278–400 km inland without fresh
deposition surfaces (Fig. 3b).

**Table 1: Mean and standard deviation (SD) of surface snow SSA for the four traverses between S16 and Dome Fuji.**

| Region[a] | Elevation (m) | Mean and SD of SSA (m$^2$ kg$^{-1}$) | | | | |
|---|---|---|---|---|---|---|
| | | 1st traverse | 2nd traverse | 3rd traverse | 4th traverse | All traverses |
| Coastal region | 615–2000 | 29 ± 15 | 24 ± 3 | 20 ± 6 | 23 ± 4 | 23 ± 8 |
| Lower katabatic wind region | 2000–3000 | 26 ± 11 | 26 ± 9 | 32 ± 24 | 26 ± 7 | 27 ± 14 |
| Upper katabatic wind region | 3000–3600 | 34 ± 9 | 37 ± 13 | 40 ± 11 | 29 ± 7 | 34 ± 11 |
| Inland plateau region[b] | 3600–3800 | 46 ± 8 | 46 ± 9 | 49 ± 13 | 39 ± 5 | 45 ± 11 |
| All regions[b] | 615–3800 | 32 ± 13 | 34 ± 13 | 36 ± 19 | 28 ± 8 | 32 ± 14 |

[a] The division of the route follows Furukawa et al. (1996). [b] Data during the stay at Dome Fuji Station (6–16 January 2022)
are excluded to avoid a bias toward the site.







**Figure 3: Spatiotemporal variation of surface snow SSA along the traverse route between S16 and Dome Fuji (DF). (a) Surface elevation based on the CryoSat-2-derived elevation model referenced to WGS84 (Helm et al., 2014). (b) Surface snow SSA measured during the first traverse. Circles indicate SSA for 10 different surfaces at each observation site, with colors representing surface morphologies. Diamonds indicate the 10-surface mean SSA. The right axis indicates $r_{eff}$, inversely proportional to the SSA on the left axis (Eq. 1). The horizontal arrow indicates the traverse direction. (c) Weather conditions observed during the first traverse. Symbols indicate cloud cover or the presence of snowfall or snow drifting, respectively. Red and green crosses indicate air temperature and wind speed measured at 12:00–13:30 LT, respectively, with the observation date noted. Arrows at the bottom indicate major sites along the traverse route. (d, e) (f, g) (h, i) (j, k) The same as (b, c) but for the second traverse, third traverse, fourth traverse, and NMD route, respectively. A, B, and C in (b), (d), and (f) indicate meteorological events whose ERA5 meteorological fields in DML are presented in Fig. S2.**

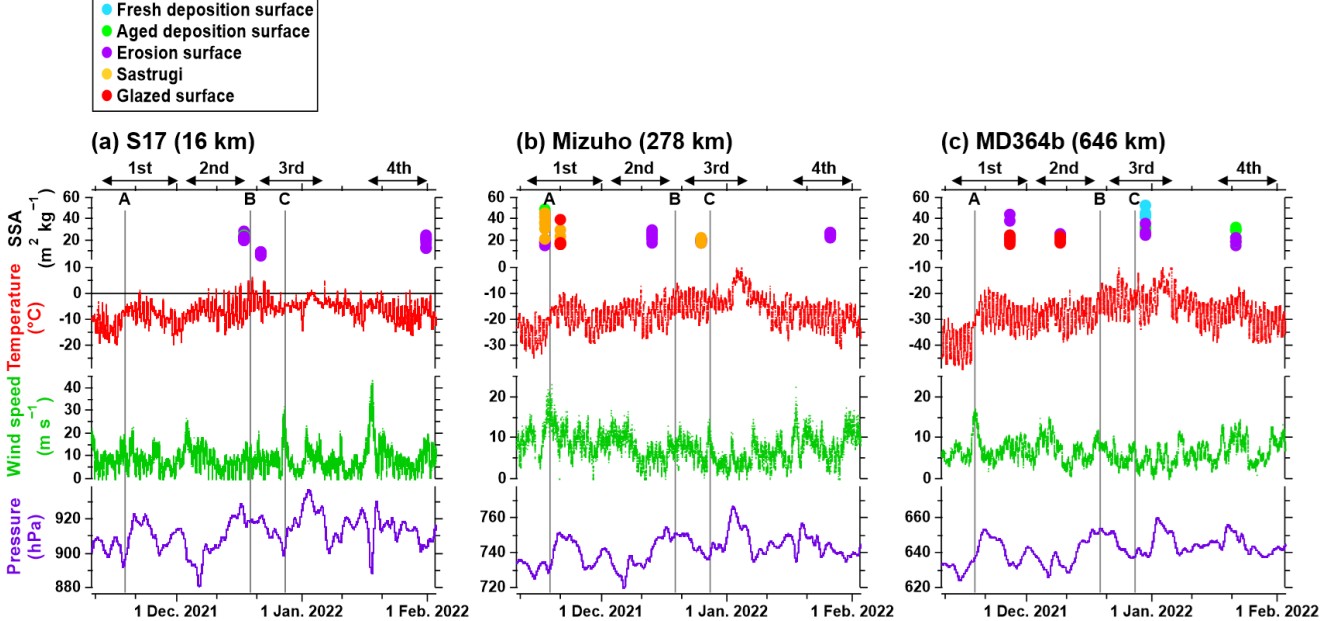

**Figure 4: Air temperature, wind speed, and air pressure between 12 November 2021 and 31 January 2022 recorded at (a) S17, (b) Mizuho, and (c) MD364b AWSs. Distances from the coast are shown in parentheses after the AWS names. The double-headed arrows above each panel represent periods for the four traverses between S16 and Dome Fuji. Markers indicate surface snow SSA measured at the AWS sites (data at S16 are shown in (a)) with colors representing surface morphologies. The vertical lines and capital alphabets A, B, and C at the top of each panel indicate meteorological events whose ERA5 meteorological fields in DML are presented in Fig. S2.**





### 3.1.2 Second traverse

Similar to the first traverse, surface snow SSA is stable from 0 to 470 km and increases beyond this range (Fig. 3d and Table 1). Erosion surfaces (aged deposition surfaces) are predominantly observed in the coastal region (inland plateau region), while glazed surfaces with low SSA are observed in the katabatic wind region (Fig. 3d). A blizzard accompanied by snowfall

occurred between 800 and 900 km, but no pronounced snow deposition was observed at wind speeds of ~ 8 m s$^{-1}$ (Fig. 3d and 3e). In contrast, a few fresh or aged deposition surfaces with a SSA of approximately 50–80 m$^2$ kg$^{-1}$ are observed between 370–660 and 940–980 km at wind speeds of 2–7 m s$^{-1}$ even in the absence of snowfall. The appearance of various surface morphologies between 380–680 km, including fresh and aged deposition surfaces and glazed surface, results in the significant variability of SSA in the range.


### 3.1.3 Third traverse

Three distinct variations in surface snow SSA, which coincide with short-term meteorological events, characterize the third traverse.

First, surface snow SSA for 10 different surfaces at S16 decreased from 19–29 m$^2$ kg$^{-1}$ on 17 December (marked by "B" in Fig. 3d) to 5–9 m$^2$ kg$^{-1}$ on 21 December (Fig. 3f). The weather remained clear skies throughout 17–21 December under high atmospheric pressure (Fig. S2), and daily maximum air temperatures reached 6°C (Fig. 4a). During the period, melt-freeze crusts appeared at the surface (see Fig. S4 for photographs of the surface and snow grains). The distinct low SSA was not observed at subsequent observation sites located more than 34 km inland, where SSA remained stable within the range of 10–

30 m$^2$ kg$^{-1}$ toward 450 km.

Second, surface snow SSA rapidly increases from 15–30 to 60–110 m$^2$ kg$^{-1}$ during the traverse between 450–480 km, with surface morphologies turning into fresh deposition surfaces of precipitation particles (marked by "C" in Fig. 3f). Beyond this area, SSA gradually decreases to 35–55 m$^2$ kg$^{-1}$ at 646 km (MD364), with precipitation particles continuing to appear on the

surface. During the period of the rapid SSA increase between 450–480 km (from the afternoon of 27 December to the morning of 28 December), heavy snowfall occurred at wind speeds below 5 m s$^{-1}$, and approximately 0.01 m thick (visual observation) snow was deposited over the entire surface. The wind speed remained below 5 m s$^{-1}$ on 28 and 29 December (Fig. 3g). The heavy snowfall under calm wind conditions is associated with water vapor advection from low latitudes to the east side of a coastal cyclone (Fig. S2), gradually reducing wind speeds toward the interior (Figs. 4 and S3).


Finally, surface snow SSA between 710–1020 km shows high variability, with 43 % of the observed surfaces being either fresh or aged deposition surfaces with SSA exceeding 50 m$^2$ kg$^{-1}$ (Fig. 3f). In this area, snowfall occurred between 770–830



km throughout the day on 2 January, with wind speeds of ~ 10 m s⁻¹ (Fig. 3g), and snow was deposited heterogeneously in patches (Fig. 3f).


### 3.1.4 Fourth traverse

The fourth traverse is characterized by a general increase in surface snow SSA toward the interior, with lower variability than the preceding three traverses (Fig. 3h and Table 1). The fresh and aged deposition surfaces with SSAs exceeding 50 m² kg⁻¹ observed between 710–1020 km during the third traverse, 15–30 days earlier (see Fig. 3f), are no longer present. The

predominant sastrugi observed between 260–450 km during the third traverse are also not observed, with aged deposition surfaces becoming predominant in the area. Moreover, the SSA values of melt-freeze crusts (5–9 m² kg⁻¹) observed at S16 on 21 December during the third traverse (Fig. 3f) are surpassed by those on 31 January (11–24 m² kg⁻¹). During the fourth traverse, the weather was primarily clear skies, and no distinct snowfall was observed (Fig. 3i).

During the fourth traverse, surface snow SSA measurements were taken by two observers individually between 278–932 km. The 10-surface mean SSA measured along different transects, separated by several tens to hundreds of meters, between 687– 932 km shows a good agreement (two lines in Fig. 3h), demonstrating the representativeness of the mean SSA along a 20 m transect for an area of hundreds of meters extent. Surface snow SSA along the NMD route between Mizuho and MD364 (26 ± 8 m² kg⁻¹) is similar to that along the MD route (26 ± 7 m² kg⁻¹) (Fig. 3h and 3j). They show no significant increase toward

the interior (slopes of linear regression are both −0.005 ± 0.013 m² kg⁻¹ km⁻¹). The frequencies for the appearance of the five surface morphologies for the two routes are also similar, with aged deposition surfaces predominantly observed.

### 3.1.5 Stay at Dome Fuji

Figure 5 shows the time series of surface snow SSA measured near Dome Fuji Station from 5 to 17 January 2022, along with

air temperature and wind speed records from Dome Fuji AWS. Surface snow SSA on 5 and 6 January ranged from 38 to 66 m² kg⁻¹, with fresh deposition surfaces observed at 30 % of the surfaces. On the morning of 6 January, surface snow SSA increased by 5 m² kg⁻¹ on average from the previous evening, with surface hoars developing on the entire surface, followed by a SSA decrease of 8 m² kg⁻¹ by the evening. During the night of 6–7 January, surface snow SSA increased to about 70 m² kg⁻¹, with heavy diamond dust occurring and approximately 5 mm thick (visual observation) snow deposited (see Fig. S5 for

photographs of surface snow crystals). On the afternoon of 8 January, fresh deposition surfaces of diamond dust were not observed at all surfaces, with 30 % of the surfaces turning into aged deposition surfaces with a low SSA of 30 m² kg⁻¹. Wind speeds on 8 January were relatively strong, reaching 7 m s⁻¹ at most (Fig. 5c). Diamond dust occurred again during the night of 8–9 January, resulting in a slight snow deposition (approximately 1 mm). From 9 to 18 January, the weather remained clear skies throughout, with daily air temperature generally decreasing (Fig. 5b). During this period, surface snow SSA generally





decreased, except for the night of 13–14 January, when SSA increased by 10 m² kg⁻¹ with no noticeable changes in surface conditions. The overall SSA decrease is accompanied by diurnal fluctuations, approximately 4 m² kg⁻¹ higher around 8:00 LT than around 20:00 LT. This phenomenon has not been reported in previous observations where measurement or data-retrieval intervals exceeded one day (e.g., Libois et al., 2015).

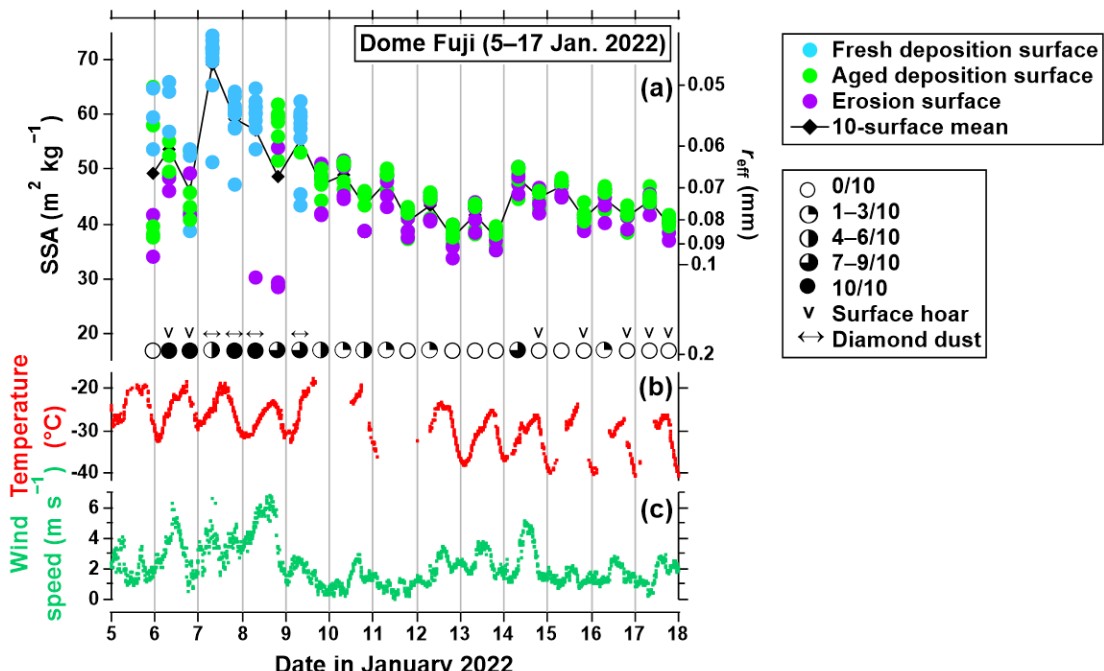


**Figure 5: Time series of surface snow SSA near Dome Fuji Station from 5 to 17 January 2022. (a) Surface snow SSA measured twice daily around 8:00 and 20:00 LT. Circles indicate SSA at 10 different surfaces with colors representing surface morphologies. Diamonds indicate the 10-surface mean SSA. Symbols at the bottom in (a) indicate cloud cover or the presence of snowfall. (b, c) Quality-controlled air temperature and wind speed data at Dome Fuji AWS,**
**respectively.**

### 3.2 Comparison of the four traverses and the stay at Dome Fuji

Figure 6 compares the 10-surface mean SSA from each observation site for the four traverses between S16 and Dome Fuji, as well as during the stay at Dome Fuji. In general, no systematic differences exceeding the SD of SSA for 10 surfaces (6.5 m²
kg⁻¹, the mean of 215 observation activities) are observed for each site between the four traverses and the stay at Dome Fuji. However, some values associated with short-term meteorological events deviate from other traverses. For instance, the mean SSA following the deposition of precipitation particles are higher, as observed at 94 km during the first traverse, 490–590 km during the second traverse, 460–650 km during the third traverse, and 7 January at Dome Fuji (refer to Figs. 3 and 5). In



contrast, the mean SSA becomes low after the appearance of melt-freeze crusts on days with positive air temperatures (S16
during the third traverse). Notably, the mean SSA at 680–980 km during the fourth traverse remains consistently lower than
in the previous three traverses, without any distinct meteorological event.

The air temperature around noon shows a seasonal variation. For example, air temperatures at 15–270 km are higher during
the second and third traverse (mid- and late-December) than during the first (mid-November) and fourth (late-January)
traverses (Fig. 6b; see also Fig. 4). Air temperature between 750–1025 km is also higher during the third traverse (early-
January) than during the other traverses. On the other hand, the wind speed around noon shows no seasonality (Fig. 6c).

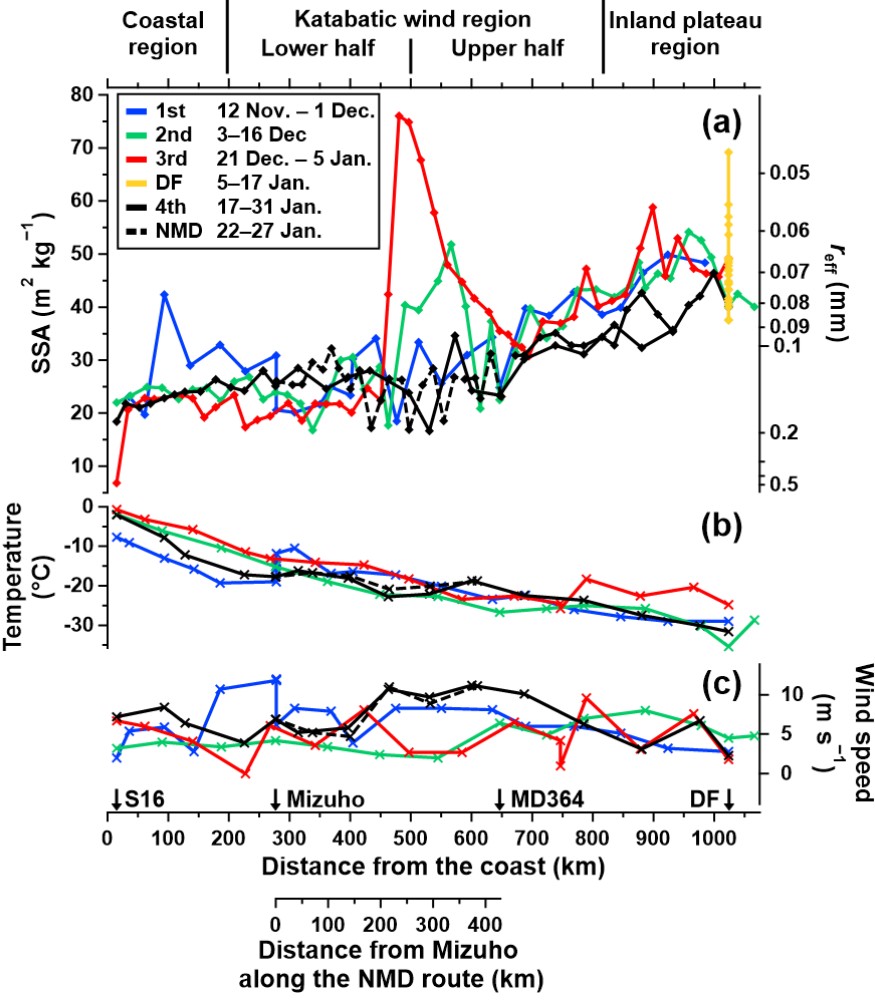

**Figure 6: In situ measured (a) 10-surface mean SSA at each observation site, (b) Air temperature around noon, and (c)
wind speed around noon during the four traverses between S16 and Dome Fuji and the stay at Dome Fuji.**



### 3.3 Compilation of the four traverses and the stay at Dome Fuji

We investigate the representative distribution of surface snow SSA in summer along the traverse route by compiling all the measured SSA data that primarily varies due to short-term meteorological events (Fig. 7a and Table 1). The SSA for all surfaces
shows a weak increase toward the interior in the coastal and lower katabatic wind regions (15–500 km), with a mean and SD of $25 \pm 9$ m$^2$ kg$^{-1}$ and a linear regression slope of $0.004 \pm 0.002$ m$^2$ kg$^{-1}$ km$^{-1}$ (excluding the data for melt-freeze crusts at S16 and precipitation particles at 460–650 km). Beyond these regions, SSA significantly increases toward the interior, reaching $45 \pm 11$ m$^2$ kg$^{-1}$ in the inland plateau region (800–1066 km), with a pronounced slope of $0.058 \pm 0.002$ m$^2$ kg$^{-1}$ km$^{-1}$.

Surface snow SSA depends on surface morphologies (Fig. 7a and Table 2). Fresh deposition surfaces show the highest SSA ($56 \pm 23$ m$^2$ kg$^{-1}$) among the five surface morphologies. Aged deposition and erosion surfaces show similar SSA in each region (Table 2) but predominantly appear in the inland plateau and coastal regions, respectively. Sastrugi, primarily observed in the lower katabatic wind region ($25 \pm 7$ m$^2$ kg$^{-1}$), shows similar SSA to erosion surfaces ($25 \pm 8$ m$^2$ kg$^{-1}$). Glazed surfaces, primarily observed in the katabatic wind region, show the lowest SSA ($19 \pm 4$ m$^2$ kg$^{-1}$) among the five surface morphologies,
resulting in higher SSA variability compared to the coastal and inland plateau regions (Table 1). The compilation of all data also reveals that surface snow SSA fluctuates at intervals of tens of kilometers in the katabatic wind region (e.g., maxima at 632 and 660 km and minima at 615 and 646 km) (Fig. 7a). These fluctuations are caused by the alternate appearance of glazed surfaces along the traverse route, which selectively occurs on steep slopes with low SMB appearing at 5–20 km intervals (vertical grey bars in Fig. 7b and 7c), as similarly observed along the same route in 1992 (Furukawa et al., 1996).






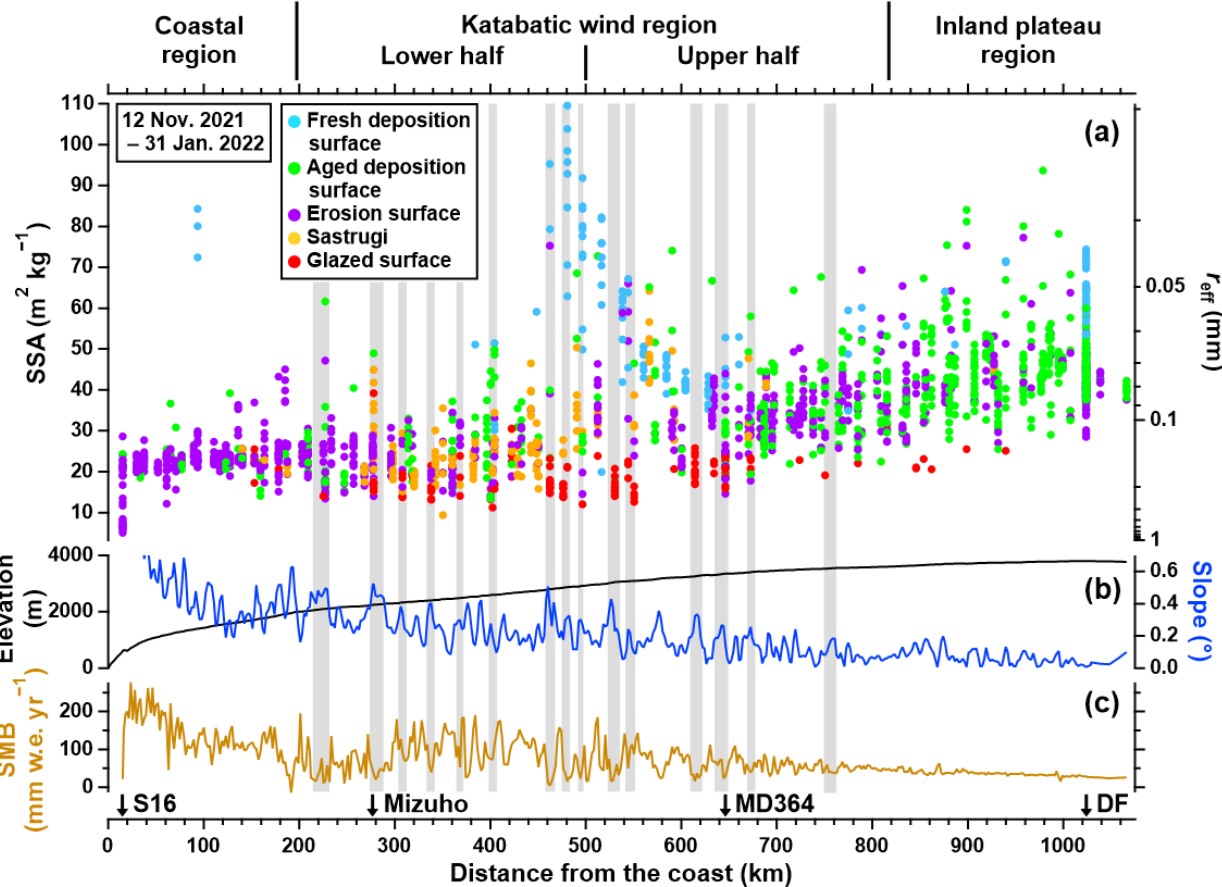

**Figure 7: Surface snow SSA, surface topography, and SMB along the traverse route between S16 and Dome Fuji. (a) Surface snow SSA for all surfaces measured during the four traverses and the stay at Dome Fuji, with the marker colors representing surface morphologies. (b) Surface elevation (black line) and slope (blue line) based on the CryoSat-2-derived elevation model referenced to WGS84 (Helm et al., 2014). (c) Mean annual SMB for 1990–2021 derived from stake measurements at 2 km intervals along the traverse route (e.g., Motoyama et al., 2015). Vertical grey bars represent sites with glazed surfaces, local maxima in surface slope, and local minima in SMB.**




**Table 2: Mean and SD of surface snow SSA for the five surface morphologies during the four traverses between S16 and Dome Fuji.**

| Region[a] | Elevation (m) | Mean and SD of SSA ($m^2 \, kg^{-1}$) | | | | |
|---|---|---|---|---|---|---|
| | | Fresh deposition surface | Aged deposition surface | Erosion surface | Sastrugi | Glazed surface |
| Coastal region | 615–2000 | 79 ± 6 | 23 ± 5 | 23 ± 6 | 23 ± 2 | 22 ± 3 |
| Lower katabatic wind region | 2000–3000 | 72 ± 21 | 28 ± 10 | 25 ± 8 | 25 ± 7 | 17 ± 5 |
| Upper katabatic wind region | 3000–3600 | 47 ± 8 | 35 ± 10 | 33 ± 9 | 39 ± 11 | 20 ± 4 |
| Inland plateau region[b] | 3600–3800 | 62 ± 8 | 45 ± 10 | 43 ± 9 | No data | 23 ± 2 |
| All regions[b] | 615–3800 | 56 ± 23 | 37 ± 12 | 29 ± 10 | 27 ± 9 | 19 ± 4 |

[a] The division of the route follows Furukawa et al. (1996). [b] Data during the stay at Dome Fuji Station (6–16 January 2022) are excluded to avoid a bias toward the site.

## 4 Discussion

Our data show little trend in SSA from 15 to 500 km from the coast, followed by a pronounced increase toward the interior from 500 to 1066 km. This is accompanied by significant variations due to the deposition of precipitation particles and the appearance of glazed surfaces (Figs. 3, 6a, and 7a). We discuss the key processes and environmental factors determining the observed spatiotemporal SSA variations between the coast and Dome Fuji.

### 4.1 Temperature dependence of snow metamorphism

Snow metamorphism is a fundamental process in SSA decrease, whose decay rate increases with snow temperature (Legagneux and Domine, 2005; Taillandier et al., 2007). The temperature dependence of snow metamorphism may produce seasonal and spatial variations in surface snow SSA along the traverse route.

High summer temperatures, either of snow or air, accelerate snow metamorphism, which can lead to a seasonal minimum in surface snow SSA, as observed at Dome C in late January (Libois et al., 2015). This seasonal temperature variation may explain the consistently lower SSA observed at 680–980 km on 18–20 January during the fourth traverse, compared to the other three traverses (Fig. 6a). However, despite the seasonal air temperature variation (Figs. 4 and 6b), no significant differences in SSA were detected for most sites between the four traverses and the stay at Dome Fuji (Fig. 6a). Thus, it appears that the seasonal



SSA variation during our observation period is masked by the short-term variations due to meteorological events, such as snowfalls and strong winds (e.g., events C and A in Fig. 3f and 3b, respectively). Below, we focus on the relationship between the spatial variation of surface snow SSA and air temperature.

Figure 8 compares the 10-surface mean SSA at each observation site (Fig. 6a) and air temperature around noon linearly
interpolated between the available data along the traverse route (Fig. 6b). They show a significant negative correlation ($r = -0.59$). Also, the observed SSA depends non-linearly on air temperature, particularly pronounced at lower air temperatures (~ −35 to −15°C) and weaker (or absent) at higher air temperatures (~ −15 to 0°C). During the third traverse, the SSA shows a gap at about −16°C and stays significantly higher than during the other traverses until −25°C. The high SSA values are caused by precipitation particles (460–650 km, Fig. 6a), which had not undergone significant snow metamorphism before the
measurements. The high SSA values are also observed when the air temperatures are high at the time of the measurements (750–1025 km, Fig. 6b), but it may be primarily influenced by low temperatures over the preceding several days, suppressing snow metamorphism. Another outlier at the bottom right of Fig. 8 represents the SSA of melt-freeze crusts observed at S16. This distinctively low SSA probably reflects the fact that the rate of snow metamorphism in the presence of liquid water differs significantly from that in the inland dry snow zone.


We discuss whether the temperature dependence of dry snow metamorphism can explain the observed non-linear relationship between air temperature and SSA. The curves in Fig. 8 are modeled SSA of snow after undergoing metamorphism for 3, 10, 30, and 50 days. These were calculated using an empirical SSA decay formula for temperature gradient metamorphism (Taillandier et al., 2007), with arbitrary temperatures and an initial SSA of 90 $m^2\,kg^{-1}$ (the mean SSA of precipitation particles
at event C, Fig. 3f). The model offers a robust depiction of the temperature dependence of SSA for metamorphosed snow, although accurately assessing the duration of snow metamorphism for the observed surfaces based on the model is difficult because the model assumes constant temperature while the observed SSA results from varying temperatures. The modeled temperature dependence is almost linear, with linear regression slopes for the 3- and 50-day curves between −35 and 0 °C being −1.05 and −0.27 $m^2\,kg^{-1}\,°C^{-1}$, respectively. The range of these slopes covers that derived from all the 10-surface mean
SSA data, −0.95 ± 0.10 $m^2\,kg^{-1}\,°C^{-1}$ (red line in Fig. 8), suggesting a primary role of air (or snow) temperature in controlling the spatial variation of surface snow SSA along the traverse route. However, the modeled linear temperature dependence does not explain the observed non-linear relationship between air temperature and SSA. Therefore, additional factors must also influence the spatial variation of SSA. These are discussed in the following sections.





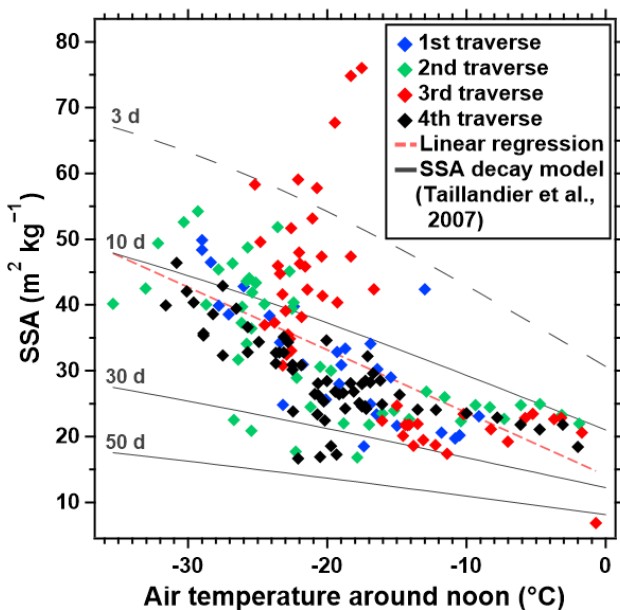


**Figure 8: Relationship between the 10-surface mean SSA at each observation site and air temperature around noon measured during the four traverses. Air temperatures for sites without measurement around noon are interpolated between the available data along the distance from the coast (Fig. 6b). The dashed red line indicates the linear regression for the 10-surface mean SSA. The curves indicate SSA of snow metamorphosed for 3, 10, 30, and 50 days, calculated**
**using an empirical SSA decay model (Taillandier et al., 2007), with an initial SSA of 90 m² kg⁻¹.**

## 4.2 Snowfall frequency

Frequent snowfall in the coastal region (e.g., Souverijns et al., 2018; Turner et al., 2019) may maintain high surface snow SSA by burying surface snow with precipitation particles more frequently than in more interior regions. This may explain the similar
SSA between 15–500 km from the coast (Fig. 6a), despite an expected decrease in SSA closer to the coast, which is anticipated due to the snow metamorphism at warmer temperatures (as shown in the range of −15 to 0°C in Fig. 8).

Satellite observations using cloud-profiling radar during 2006–2011 (Palerme et al., 2014) indicate snowfall frequencies of 20–30 % (fraction of observation time) at 0–200 km, 10–20 % at 200–500 km, and < 10 % at 500–1066 km along our traverse
route. These frequencies can be interpreted as indicative of the rate at which precipitation particles bury the surface and can be used to estimate the relative duration of exposure (or metamorphism) for specific snow layers at the surface. This means that the duration of snow metamorphism decreases toward the coast to about one-third of that observed at 500 km inland. According to the SSA decay model (Taillandier et al., 2007), this reduced duration of snow metamorphism results in a SSA increase of approximately 10 m² kg⁻¹ near the coast (see curves at ∼ −10 to 0 °C in Fig. 8; see also Fig. 6b ) relative to that at





500 km inland. This relative SSA increase may adequately compensate for the expected SSA decrease from 500 km inland to the coast due to the temperature dependence of snow metamorphism (e.g., −9 to −16 m$^2$ kg$^{-1}$ from −20 to 0 °C for snow SSA after 10–30 days of metamorphism, Fig. 8). Therefore, increasing snowfall frequency toward the coast likely explains the observed similar SSA between 15 and 500 km (Fig. 6a).

### 4.3 Wind-driven inhibition of snow deposition

Strong winds can blow away falling or deposited snow, prolonging the metamorphism of certain snow layers at the surface (e.g., Lenaerts et al., 2017). This wind-induced decrease in SSA is evident in the katabatic wind region, where glazed surfaces are formed by an accumulation hiatus through consistent strong winds and exhibit the lowest SSA among the five surface morphologies (see Fig. 7a and 7c). We assess a possible decrease in surface snow SSA due to the wind-driven inhibition of 520    snow deposition along the traverse route.

First, we identify wind conditions that inhibit snow deposition. During our four traverses, heterogeneous or no snow deposition despite snowfall was observed at wind speeds of approximately 5–10 m s$^{-1}$ (e.g., 94 km during the first traverse, 800–900 km during the second traverse, and 770–830 km during the third traverse, Fig. 3; 8 December at Dome Fuji, Fig. 5). Our in situ 525    weather observations also show that 93 % of snow drifting occurs at wind speeds above 5 m s$^{-1}$, and snow drifting begins at wind speeds of 5–6 m s$^{-1}$ in the presence of snowfall (Fig. 9a). This wind speed threshold aligns with the findings that saltation and creep of snow particles only occur at wind speeds above 5 m s$^{-1}$ (Kosugi et al., 1992; Filhol and Sturm, 2015).

Wind speed frequency distributions between November 2021 and January 2022, recorded at eight AWSs along the traverse 530    route, show higher wind speeds at distances of 16–646 km from the coast than at 1024 and 1066 km (Fig. 9b; refer to Fig. S3 for the data of the AWSs). The frequency of wind speeds exceeding the threshold of 5 m s$^{-1}$ notably increases from 19 % and 32 % at 1024 and 1066 km, respectively, to 69 % at 646 km inland, with a minor increase observed beyond this site toward the coast (73–79 %).

We assume the frequencies of wind speeds exceeding 5 m s$^{-1}$ as an indicative of wind-driven inhibition of snow deposition. The frequencies at 646 km (69 %) are approximately 2–4 times higher than those at 1024 and 1066 km (19 % and 32 %, respectively), potentially resulting in approximately 2–4 times longer exposures (or metamorphism) of snow layers at the surface. According to the SSA decay model (Taillandier et al., 2007), this 2–4 times longer snow metamorphism leads to a decrease in SSA by approximately 10–20 m$^2$ kg$^{-1}$ at around 646 km inland (see curves at −25 to −20 °C in Fig. 8; see also Fig. 540    6b) relative to those at around 1024–1066 km inland. This relative SSA decrease synchronizes with the SSA decrease toward the coast in response to increasing temperature in the area (−35 to −20 °C in Fig. 8), likely explaining the observed pronounced decrease in SSA from 1066 to 500 km from the coast (Fig. 6a).



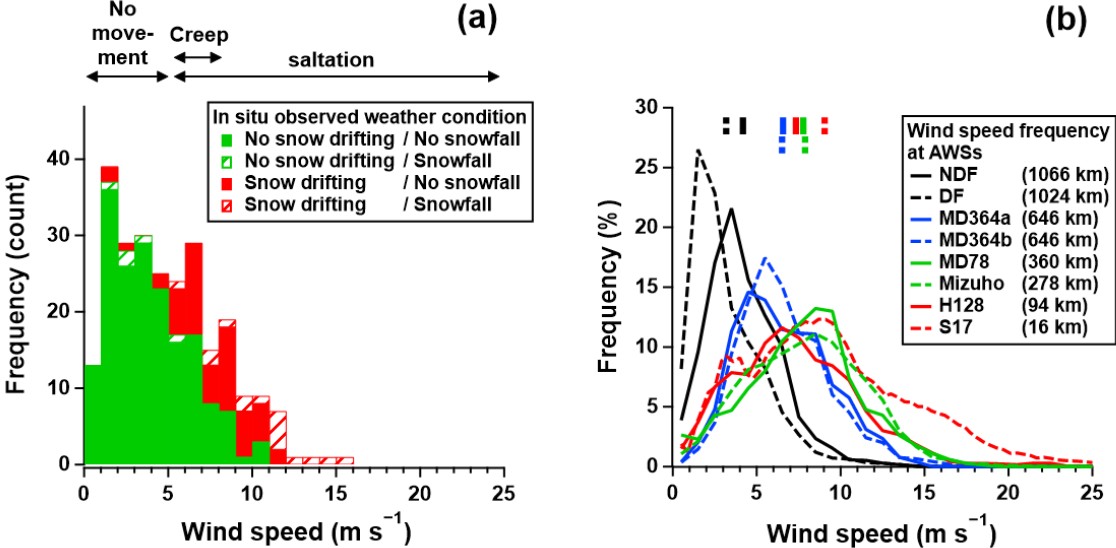

**Figure 9: (a) Frequency distribution of wind speed measured three times daily (6:00–7:00, 12:00–13:30, and 19:00–**
**20:30 LT) during the four traverses. The wind speed bin size is 1 m s⁻¹. Green (red) columns indicate wind speed**
**frequency in the absence (presence) of snow drifting. Solid (striped) columns indicate wind speed frequency in the**
**absence (presence) of snowfall. The double-headed arrows above the panel indicate snow particle motion depending on**
**wind speed (Filhol and Sturm, 2015). (b) Frequency distribution of wind speed between November 2021 and January**
**2022 recorded at eight AWSs installed along the traverse route (Fig. S3). Vertical lines at the top of the panel indicate**
**mean wind speeds for the period. Distances from the coast at the AWSs are shown in parentheses in the legend.**

### 4.4 Surface hoar formation

The formation of surface hoars may influence the spatial variation of surface snow SSA by playing a role similar to snowfall
(Domine et al., 2007; Gallet et al., 2014). Surface hoar is typically formed in the inland plateau region under calm wind
conditions, possibly contributing to the high SSA in the region (see Figs. 6a and 7a). Indeed, we observed an increase in surface
snow SSA after surface hoar was well-developed on 6 January at Dome Fuji (Fig. 5a). Increases in SSA by approximately 4
$m^2 kg^{-1}$ during each night (20:00–8:00 LT) from 9 to 17 January (Fig. 5a) may also be associated with surface hoar formation
(e.g., Gallet et al., 2014). Although surface hoar can sublimate under unsaturated air conditions, the cumulative daily increases
in SSA may be significant. Quantifying the net contribution of surface hoar formation and sublimation to surface snow SSA
requires more detailed observations on diurnal changes in surface snow SSA and its relationship to the presence of surface
hoar.



## 4.5 Other factors influencing the spatial variation of SSA

The potentially high initial SSA in the inland plateau region, resulting from diamond dust (Walden et al., 2003) or wind-driven
fragmentation of snow grains (Gallet et al., 2011), may contribute to the high surface snow SSA in the region. For example,
the snow that has metamorphosed at temperatures ranging from −30 to −10 °C over 10 days from an initial SSA of 150 m$^2$
kg$^{-1}$, close to the highest value for precipitation particles of needles and columns (Domine et al., 2007), keeps a higher SSA
by 8–9 m$^2$ kg$^{-1}$ than that starting from a SSA of 90 m$^2$ kg$^{-1}$ (Taillandier et al., 2007). However, assessing the effect of initial
SSA on the spatial variation of surface snow SSA requires understanding the spatial variation in the SSA of freshly deposited
snow over Antarctica.

Wind-driven sublimation and condensation in snow (e.g., Albert, 2002; Ebner et al., 2016) may facilitate snow metamorphism,
particularly in the coastal and katabatic wind regions (see Fig. 9b). Additionally, the temperature gradient in the top few
centimeters, which is not parameterized in the SSA decay model (Taillandier et al., 2007), may vary along the traverse route
and produce differences in the SSA decay rate. For example, the temperature gradient possibly increases toward the interior
due to increasing diurnal air temperature variations (see Figs. 4 and S3) or decreasing wind speed that diffuses heat within the
snow (Fig. 9b), which may facilitate snow metamorphism more in the inland plateau region than in the katabatic wind and
coastal regions. Assessing the impact of these factors on the spatial variation of surface snow SSA requires further quantitative
understanding of the relationship between the wind speeds (or temperature gradients) and SSA decay rate.


## 5 Conclusions

We measured surface snow SSA using HISSGraS during two round-trip traverses – four traverses on the same path – between
the coast near Syowa Station and Dome Fuji from November 2021 to January 2022. Quick SSA measurements using HISSGraS,
which directly measures snow surface without requiring sampling, enabled us to collect 215 sets of SSA data for 10 different
surfaces along a 20 m transect. Our data provide the first detailed view of the wide-area distribution of surface snow SSA in
Antarctica based on ground-based observations, featuring high spatial observation intervals (approximately 5 km between
adjacent observation sites).

Surface snow SSA shows no elevation or temperature dependence between 15 and 500 km from the coast (elevation: 615–
3000 m) along the traverse route, with a mean and SD of 25 ± 9 m$^2$ kg$^{-1}$. Beyond this range, SSA increases toward the interior,
reaching 45 ± 11 m$^2$ kg$^{-1}$ between 800 and 1066 km (3600–3800 m). SSA dynamically fluctuates depending on surface
morphologies and short-term meteorological events associated with offshore cyclone activities or its blockage by high-pressure
ridges. For example, (i) Glazed surfaces, formed by an accumulation hiatus at intervals of tens of kilometers in the katabatic
wind region, exhibit low SSA (19 ± 4 m$^2$ kg$^{-1}$), reducing the mean SSA and increasing SSA variability. (ii) Freshly deposited



snow shows high SSA (60–110 m² kg⁻¹). However, the snow deposition is inhibited by wind-driven snow drifting at wind speeds above 5 m s⁻¹, resulting in heterogeneous or no snow deposition. Wind speeds reaching 20 m s⁻¹ even erode the surface, exposing aged snow with low SSA. (iii) The appearance of melt-freeze crusts decreases surface snow SSA to 5–9 m² kg⁻¹, when daily maximum air temperatures become positive during continuous clear sky days under high atmospheric pressure.

We discussed the key processes and environmental factors determining the observed spatial variation of surface snow SSA. The observed SSA is negatively correlated with air temperature and characterized by a non-linear dependence on air temperature; it is weaker (or absent) at higher air temperatures between 15–500 km and pronounced at lower air temperatures between 500–1066 km. While the overall temperature dependence of the observed SSA is consistent with the range of modeled temperature dependence of snow SSA metamorphosed over tens of days, the observed non-linearity in the temperature 605 dependence cannot be explained by the modeled linear temperature dependence. The weak correlation of observed SSA with temperature between 15–500 km may be explained by an increasing snowfall frequency toward the coast, which maintains high surface snow SSA near the coast by frequently burying surface snow with precipitation particles. The pronounced dependence of SSA on temperature between 500–1066 km may be explained by an increasing frequency of wind speeds exceeding 5 m s⁻¹ toward the coast within the area, which inhibits snow deposition by frequent snow drifting and prolongs the 610 metamorphism of snow layers at the surface closer to 500 km. Overall, these findings emphasize the crucial roles of temperature-dependent snow metamorphism, snowfall frequency, and wind-driven inhibition of snow deposition in the spatial variation of surface snow SSA in the Antarctic inland.

Future research should explore additional factors to gain a more comprehensive understanding of the spatial variation of surface 615 snow SSA. These factors include surface hoar formations, which may contribute to high SSA in the inland plateau region. Understanding the spatial variability in the initial SSA of freshly deposited snow over Antarctica might also be necessary. Additionally, assessing how snow metamorphism is facilitated by wind-driven sublimation and condensation, as well as by large temperature gradients in the top few centimeters of snow, would be desirable.

Our dataset provides abundant ground-truth SSA data for validating satellite-derived and model-simulated SSA variations across Antarctica. Our insights into the crucial processes controlling the spatial variation of surface snow SSA will contribute to improving the parameterization of snow SSA in climate models (e.g., Flanner and Zender, 2006), thereby better constraining present and future changes in surface albedo in Antarctica.



**Data availability**

All data observed in this study are available at the NIPR ADS data repository (https://ads.nipr.ac.jp/dataset/A20240308-001, URL during revision).

**Author contribution**

RI, TA, SF, and KK designed the field observation. RI performed the SSA observation during the four traverses with support from ST, HM, and FN. RI processed and analyzed all the data and wrote the manuscript with inputs from all the other authors. All authors contributed to the discussion and reviewed the manuscript.

**Competing interests**

The contact author has declared that none of the authors has any competing interests.

**Acknowledgments**

Field campaigns were conducted as part of the Japanese Antarctic Research Expedition (JARE), supported by the National Institute of Polar Research (NIPR) under the Ministry of Education, Culture, Sports, Science and Technology (MEXT). We thank all participants in the fieldwork who contributed to field logistics and maintaining AWSs. We thank Rei Niimi (Japan Meteorological Agency) and Shohei Morino (Nagoya University) for their supports in in situ weather observations, Naohiko Hirasawa (NIPR) for helping us obtain and interpret meteorological data from AWSs and the ERA5 reanalysis, and the Japan Meteorological Agency for providing us with the S17 AWS data.

**Financial support**

This study has been supported by the Japan Society for the Promotion of Science and MEXT KAKENHI (grant no. 18H05294 to Shuji Fujita).



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
