# Peer review of "Spatial variation in the specific surface area of surface snow measured along the traverse route from the coast to Dome Fuji, Antarctica, during austral summer"

_EGUsphere, 2024_

## Author Comment (AC1)

The authors would like to thank Martin Schneebeli for acknowledging the value of our study and providing valuable feedback on our manuscript. We addressed all of your comments. Our responses are provided in blue, and the revised parts of the manuscript are marked in red.

Reviewer #1: Martin Schneebeli

5    The authors describe and discuss detailed measurements of surface snow SSA in East Antarctica. The methods are described in detail and reproducible. This dataset may serve as ground truth validation for optical remote sensing. The sensor measures the surface (a few millimetres of the snowpack). The repeat measurements on the 1000 km traverses with very high spatial resolution illustrate the complexity of the snow surface in Antarctica.

10    The surface reflectivity is certainly the most important factor in albedo calculations. However, the very small penetration depth of light at 1320 nm into snow is also a limitation in the albedo calculation, as the visible and short near-infrared wavelength backscattering also occurs deeper into the snowpack. This should be clarified in more detail in this paper and discussed.

[A01] We will clarify the limitation in albedo calculations, by estimating uncertainties in broadband
15    albedo calculated using a physically based snow albedo model (Aoki et al., 2011). The estimated uncertainty will be mentioned in Sect. 2.2.1 (see below). Since albedo calculations are beyond the scope of this study, the detailed scheme of the calculation will be described and discussed in the Supplementary Note (see below).

20    Sect. 2.2.1

"*We used HISSGraS for SSA measurements, which shares* a similar *measurement principle as IceCube (Gallet et al., 2009) but offers advantages such as being lightweight, handheld, and capable of directly measuring snow surfaces without the need for sampling (Aoki et al., 2023). It employs an integrating sphere, the circular part of which (25 mm diameter) is a glass window. Inside, a laser diode and an*
25    *InGaAs photodiode are attached. The laser diode emits NIR light at 1310 nm through the glass window, which is in direct contact with the snow surface, and*
30     *the InGaAs photodiode collects the light reflected by the snow surface. The measured light intensity is then converted to reflectance (R) using a calibration curve derived from the measurements on six reflectance standards (5–99%). Since the calibration curve varies with ambient temperature due to the temperature sensitivity of the laser diode emission* $(-1\%\ K^{-1})$*, HISSGraS records*
35    *the temperature close to the laser diode for every light intensity measurement, enabling the correction for the temperature dependence of calibration curves. Following Aoki et al. (2023), we constructed a*

*calibration formula applicable to the temperature range observed during our study (−35 to 5°C) (see Supplementary Note S1 and Fig. S1 for details). Finally, the calibrated R is converted to SSA using a theoretical R–SSA relationship derived from a radiative transfer model that assumes spherical snow grains and employs Mie theory (Aoki et al., 1999).*

     *The penetration depth – the depth at which the light intensity reduces to $e^{-1}$ of its incident value – for NIR light at 1310 nm is approximately 8 mm for fresh snow and 9 mm for Antarctic depth hoar with a SSA of 40 and 12 $m^2$ $kg^{-1}$ and a density of 120 and 230 $kg$ $m^{-3}$, respectively (Gallet et al., 2011). Therefore, HISSGraS provides a weighted average of the snow SSA over approximately the top 10 mm of near-surface snow (referred to hereafter as "surface snow SSA"). The SSA measured with HISSGraS for the depths enables broadband albedo calculations with an uncertainty of 0.03 in the Antarctic inland, despite the deeper penetration of visible and short near-infrared wavelengths into the snowpack, according to a physically based snow albedo model (see Supplementary Note S2 and Fig. S2)."*

**Supplementary Note S2**

*"Note S2: Potential error in broadband albedo derived from HISSGraS measurement*
*To evaluate the potential error in broadband albedo derived from the surface snow grain size measured with HISSGraS, we compared the albedo values calculated for a snow layer with homogeneous snow grain size to those calculated when the snow grain size of the subsurface layer differs from the surface layer. Broadband albedos were calculated using a physically based snow albedo model (Aoki et al., 2011), which assumes two snow layers. The snow grain shape employed was a spherical particle model, with radii (SSA) in the two snow layers set at 50 μm (65.4 $m^2$ $kg^{-1}$), 200 μm (16.4 $m^2$ $kg^{-1}$), and 1000 μm (3.3 $m^2$ $kg^{-1}$), representing average snow grain radii for new snow, fine-grained old snow, and old snow near the melting point, respectively (Wiscombe and Warren, 1980). The thickness of the top layer was assumed to be the critical snow depth (CSD) for monochromatic albedo at 1310 nm, the laser wavelength of the HISSGraS. The CSD is defined as the depth at which the monochromatic albedo at this wavelength closely approximates (assumed to be 99% in this study) the albedo of a semi-infinitely thick snow layer and no longer depends on the snow grain size in deeper layers. For a snow density of 200 $kg$ $m^{-3}$, the CSD values are 8.3 mm, 21.1 mm, and 37.1 mm for snow grain radii ($r_s$) of 50, 200, and 1000 μm, respectively. The thickness of the bottom layer in the two-layer snow model was assumed to be semi-infinite. Snow contamination is held constant at a concentration of 1.0 ng $L^{-1}$ black carbon across all snow layers, based on in-situ measurements along the route from S16 to Mizuho (Kinase et al., 2020). The atmospheric conditions include assumptions of both clear sky and cloudy sky. Considering that the solar zenith angle ($θ_0$) at local solar noon on the summer solstice is 44.5° and 54.0° at S16 and Dome Fuji, respectively, we calculated the broadband albedos for $θ_0 > 45°$.*

     *Figure S2 presents the simulated broadband albedo under clear sky (Fig. S2a) and cloudy sky (Fig. S2b) conditions for various combinations of three types of $r_s$ across the two snow layers. When the $r_s$ values in the two snow layers are the same and correspond to the HISSGraS measurement, the simulated broadband albedo corresponds with that expected from the HISSGraS measurement. When the $r_s$ values differ between the top and bottom layers, the potential variability in broadband albedos for each $r_s$ in the top layer is calculated. The difference in albedo between cases where the snow grain size is the same and differs between the two snow layers represents a potential error in the broadband albedo estimated from the surface snow grain size measured with the HISSGraS. The maximum estimated error is 0.05 when the*

*$r_s$ of the top and bottom layers are 1000 μm and 50 μm, respectively, at $θ_0 = 45°$ under clear sky*
80  *conditions. This error remains consistent across all $θ_0$ under cloudy conditions. In this study, the surface*
*snow SSA measured with HISSGraS in Antarctica predominantly falls within the range of 50 μm to 200*
*μm (Fig. 7). For this range, the estimated maximum error is 0.03 when the $r_s$ for the top and bottom*
*layers are 50 μm and 1000 μm, respectively, under the same $θ_0$ conditions.*"

[Figure]

85  ***Figure S2: Theoretically calculated broadband albedos under (a) clear sky and (b) cloudy sky for the***
***combination of $r_s$ = 50, 200, and 1000 μm in the two-layer snow model as a function of $θ_0$ simulated***
***with a physically based snow albedo model.***

Finally, we will note the necessity of SSA data at deeper depths for more accurate albedo calculations in
L621–623 in the original manuscript: "*Our insights into the crucial processes controlling the spatial*
90  *variation of surface snow SSA will contribute to improving the parameterization of snow SSA in climate*
*models (e.g., Flanner and Zender, 2006). Further investigation of SSA at deeper depths in the top few*
*tens of centimeters, along with snow grain shape analysis for the calculations of bidirectional reflectance*
*distribution function necessary for satellite albedo retrievals (e.g., Ishimoto et al., 2018; Robledano et*
*al., 2023), would be desirable for better constraining present and future changes in surface albedo in*
95  *Antarctica.*"

I was also missing measurements of any snow impurities. Known to be small, they may not be negligible,
especially in the coastal region, and could contribute to an altered albedo.
[A02] Recently, the mass concentration of black carbon, a significant light-absorbing impurity, in the
snow has been measured along a traverse route from Syowa Station to Mizuho, showing a maximum of

100      1.2×10$^{-3}$ ppm (Kinase et al., 2020). According to broadband albedo simulations as a function of black carbon, this concentration level has little effect on albedo (e.g., see Fig. 3 in Aoki et al., 2011).

        We will cite these studies and modify L36 in the original manuscript to "*For example, the SSA is an important snow physical parameter for surface albedo. Near-infrared albedo strongly depends on snow grain size, while visible albedo is more influenced by the concentrations of light-absorbing impurities*

105      *(Warren and Wiscombe, 1980; Wiscombe and Warren, 1980; Aoki et al., 2011). In Antarctica, the impurity concentration is low enough not to affect albedo (Grenfell et al., 1994; Warren et al., 2006; Kinase et al., 2020). The snow thickness affecting albedo is the top few tens of centimeters because light penetration depth ranges from several millimeters at near-infrared wavelengths to several tens of centimeters at visible wavelengths (Zhou et al., 2003). Therefore, the SSA in the top few tens of*

110      *centimeters is a key determinant for surface albedo in Antarctica.*"

The manuscript is a very important contribution to a better understanding of the antarctic snowpack and its interaction with short-wave radiation.

[A03] Thank you for recognizing the value of this study.

For directional reflectance calculations, the snow particle shape is important. Robledano et al. 2023

115      demonstrate that snow grain shape is an important factor in BRDF-calculations, which are necessary for satellite albedo retrievals. Could the author also record the snow particle shape together with their instrumental observations?

[A04] Thank you for informing us of the related study. Unfortunately, we were unable to record the detailed snow grain shape for all observed surfaces due to time constraints (microscopic photographs of

120      snow grains were obtained for approximately 1/4 of the surfaces, and some of these were referred to identify snow grain shape).

        We will mention the importance of snow grain shape for albedo constraints in L621–623 in the original manuscript; "*Our insights into the crucial processes controlling the spatial variation of surface snow SSA will contribute to improving the parameterization of snow SSA in climate models (e.g., Flanner*

125      *and Zender, 2006). Further investigation of SSA in the top few tens of centimeters, along with snow grain shape analysis for the calculations of bidirectional reflectance distribution function necessary for satellite albedo retrievals (e.g., Ishimoto et al., 2018; Robledano et al., 2023), would be desirable for better constraining present and future changes in surface albedo in Antarctica.*"

The authors mention the potential of these measurements to be used for satellite remote sensing

130      validation. However, there are no temporally coincident optical satellite passes mentioned, so will the large spatial and temporal variability of the surface snow SSA allow later for a direct comparison?

[A05] We checked when satellites, capable of retrieving surface snow SSA, passed over the observation area. Terra passed within 6:00–10:00 LT during our traverses, and Aqua within 15:00–18:00 LT. The observations at Dome Fuji were scheduled to coincide with these times. Some observations during

135      traverses were conducted outside these times. However, since temporal changes in SSA are minimal at low temperatures (e.g., SSA decreases only from 40 m$^2$ kg$^{-1}$ to 38 m$^2$ kg$^{-1}$ over 5 hours at -10°C;

Talandier et al., 2007) except during and immediately after snowfall, SSA measurements taken a few hours before and after satellite passes allow later for a comparison with satellite retrievals.

We will modify L171 in the original text to "*Additionally, we performed the activity twice daily (around 8:00 and 20:00 LT, close to the time when Terra and Aqua satellites pass Dome Fuji) at a fixed location near Dome Fuji Station from 5 to 17 January to track the temporal variation of SSA.*"

For additional manuscript revision, see [A09] of this file.

The authors nicely illustrate the observed trends and spatial variability in Fig. 7. In table 2, it would be interesting to see if the differences between the Regions are significant.

[A06] We will modify L416–420 to "*Aged deposition and erosion surfaces show similar SSA in each region (Table 2) but predominantly appear in the inland plateau and coastal regions, respectively. Both surfaces show a significant increase in SSA toward the interior, exceeding their SDs (Table 2). Sastrugi, primarily observed in the lower katabatic wind region ($25 \pm 7$ $m^2$ $kg^{-1}$), shows similar SSA to erosion surfaces ($25 \pm 8$ $m^2$ $kg^{-1}$). Glazed surfaces, primarily observed in the katabatic wind region, show the lowest SSA ($19 \pm 4$ $m2$ $kg^{-1}$) among the five surface morphologies, with similar values within the SD across the four regions (Table 2).*"

Significance assessment for fresh deposition surfaces and sastrugi is refrained due to the limited number of observed surfaces in many of the four regions.

There is always a temperature gradient in the snowpack in Antarctica, so isothermal metamorphism is not relevant. The reviewer suggests that more weight is put on temperature gradients and not absolute temperature.

[A07] We acknowledge that there is always a temperature gradient in the top few meters of Antarctic firn. We will put more weight on temperature gradient metamorphism (not isothermal metamorphism) in Sect. 4.1 (see below). We also think that the effect of absolute temperature on temperature gradient metamorphism should not be ignored because it significantly influences the amount of saturated water vapor and, hence, the rate of snow metamorphism. Evidentially, the snow metamorphism experiments have shown the non-negligible dependence of SSA decay rate (or grain growth rate) on absolute temperature (between −20 and −4 °C) even under temperature gradient conditions between 8 and 54 °C $m^{-1}$ (Marbouty, 1980; Taillandier et al., 2007). Also, seasonal variations in surface snow SSA during summer may support this temperature dependence of the SSA decay rate (Libois et al., 2015). Thus, we believe that the temperature dependence of our SSA data is worth discussing.

Sect. 4.1
"*We discuss whether the temperature dependence of dry snow metamorphism can explain the observed non-linear relationship between air temperature and SSA. The curves in Fig. 8 are modeled SSA of snow after undergoing metamorphism for 3, 10, 30, and 50 days under temperature gradient (solid lines) and isothermal (dashed lines) conditions. These were calculated using two empirical SSA decay formulas for temperature gradients between 8 and 54 °C $m^{-1}$ and temperatures between −20 and −4 °C and for temperatures between −15 and −4 °C without gradient, respectively (Taillandier et al., 2007), as a function of temperature and an initial SSA of 90 $m^2$ $kg^{-1}$ (the mean SSA of precipitation particles at event*

*C, Fig. 3f). SSA decreases more slowly under isothermal conditions than under temperature gradient conditions, expecting approximately 50 days of snow metamorphism without burial for the observed surfaces in the higher temperature range (~ −15 to 0 °C). Such a prolonged accumulation hiatus seems unrealistic, considering frequent accumulation due to offshore cyclones near the coast of the traverse route (Takahashi et al., 1994; Watanabe, 1978). Thus, temperature gradient metamorphism may better explain SSA decrease in surface snow (the top ~10 mm) in the Antarctica inland. The model for temperature gradient conditions offers a robust depiction of the temperature dependence of SSA for metamorphosed snow, although accurately assessing the duration of snow metamorphism for the observed surfaces based on the model is still difficult because the model assumes constant temperature while the observed SSA results from varying temperatures and because it does not incorporate temperature gradient as a variable (the potential effect of temperature gradient on the spatial SSA variations is discussed in Sect. 4.5). The modeled temperature dependence is almost linear, with linear regression slopes for the 3- and 50-day curves between −35 and 0 °C being −1.05 and −0.27 $m^2$ $kg^{−1}$ $°C^{−1}$, respectively. The range of these slopes covers that derived from all the 10-surface mean SSA data, −0.95 ± 0.10 $m^2$ $kg^{−1}$ $°C^{−1}$ (red line in Fig. 8), suggesting a primary role of air (or snow) temperature in controlling the spatial variation of surface snow SSA along the traverse route. However, the modeled linear temperature dependence does not explain the observed non-linear relationship between air temperature and SSA. Therefore, additional factors must also influence the spatial variation of SSA. These are discussed in the following sections.*

[Figure]

**Figure 8: *Relationship between the 10-surface mean SSA at each observation site and air temperature around noon measured during the four traverses. Air temperatures for sites without measurement around noon are interpolated between the available data along the distance from the coast (Fig. 6b). The red line indicates the linear regression for the 10-surface mean SSA. The* grey *curves indicate* SSA of snow metamorphosed for 3, 10, 30, and 50 days under temperature gradient (solid lines) and**

*isothermal (dashed lines) conditions, calculated using an empirical SSA decay model (Taillandier et al., 2007), with an initial SSA of 90 m$^2$ kg$^{−1}$.*"

We also emphasize the need for further study on the effect of temperature gradients on the observed spatial SSA variations in Sect. 4.5 (see below). To our knowledge, quantitatively assessing the impact of the magnitude and frequency of temperature gradients on SSA decrease at the surface in Antarctica is currently challenging due to the lack of knowledge about spatiotemporal temperature gradient variations in the top ~10 mm across the traverse route, where wind ventilation and penetration of insolation into the firn complicate temperature gradients.

Sect. 4.5

"*Wind-driven sublimation and condensation in snow (e.g., Albert, 2002; Ebner et al., 2016) may facilitate snow metamorphism, particularly in the coastal and katabatic wind regions (see Fig. 9b). Additionally, the magnitude and frequency of temperature gradients in the top few centimeters, which is not parameterized in the SSA decay model (Taillandier et al., 2007), is important for snow metamorphism. In fact, the model underestimates the observed SSA decay rate during 27–29 December when SSA decreases from 60–110 to 35–55 m2 kg$^{−1}$ within 2 days at around −20°C whereas the model estimates 3–15 days for this decrease (Fig. 8). This discrepancy may arise because the actual temperature gradient within 10 mm of the surface is stronger (e.g., exceeding 100°C m$^{−1}$,(Azuma et al., 1997)) than the conditions on which the empirical model is based, suggesting essential role of large temperature gradients in spatial SSA variations. The magnitude and frequency of the temperature gradient may vary along the traverse route and produce differences in the SSA decay rate. For example, the temperature gradient possibly increases toward the interior due to increasing diurnal air temperature variations (see Figs. 4 and S3) or decreasing wind speed that diffuses heat within the snow (Fig. 9b), which may facilitate snow metamorphism more in the inland plateau region than in the katabatic wind and coastal regions. Assessing the impact of wind ventilation and temperature gradient on the spatial variation of surface snow SSA requires further quantitative understanding of the relationship between the wind speeds (or the magnitude and frequency of temperature gradients) and SSA decay rate. It is also necessary to understand temperature gradient variations in the top few centimeters across Antarctica where wind ventilation and penetration of insolation into the firn may complicate temperature gradients.*"

Title: "Spatiotemporal" implicitly suggests that the measurements are during an entire year.    I suggest as title "Spatial variation ... during austral summer"

[A08] We agree with your suggestion and will change the title to "*Spatial variation in the specific surface area of surface snow measured along the traverse route from the coast to Dome Fuji, Antarctica, during austral summer.*"

25: "enable the validation ..." this is not shown in greater detail in the paper

[A09] SSA retrievals from e.g., the MODIS sensor at 860, 1240, and 1640 nm on the Terra and Aqua

satellites can be compared with our in-situ data (particularly directly at 1240 nm) (see also [A05])

We will revise L66 in the manuscript to "*Algorithms for retrieving the SSA of near-surface snow using near-infrared (NIR) imagery data at 860, 1240, and 1640 nm, such as from the moderate resolution imaging spectroradiometer (MODIS) onboard Terra and Aqua satellites, or microwave data have been developed and applied to Antarctica*".

We will modify L620 to "*Our dataset provides abundant ground-truth SSA data for validating satellite-derived SSA variations across Antarctica, such as from Terra and Aqua MODIS data (Scambos et al., 2007; Jin et al., 2008; Kokhanovsky et al., 2011), Ocean and Land Colour Instrument (OLCI) onboard Sentinel-3A/B (Kokhanovsky et al., 2019) and Second-Generation Global Imager (SGLI) onboard Global Change Observation Mission-Climate (GCOM-C) (Hori et al., 2018).*"

Detailed validation is beyond the scope of this study.

43 ff: The isothermal case does not exist in Antarctica, and can be deleted here (or then the newer papers by Kaempfer et al, Calonne et al must be cited and commented.
[A10] We will delete the sentences related to isothermal metamorphism.

54: the sentence "In addition ..." is not relevant in the context of this paper
[A11] We will remove the sentence.

75: ASSSAP was mainly used in Antarctica: I suggest to delete "alpine"
[A12] We will remove "alpine".

176: "shares the same measurement principle ..." replace by " shares a similar measurement principle ..."
[A13] We will replace it.

203: "... with the accurate ..." there is quite a large measurement uncertainty by the methane absorption method, give precision in text
[A14] We will remove the word "accurate" and add the accuracy of 12% (Legagneux et al., 2002) in L203 in the original text.

249: You could mention here that the measurements are during austral summer, and not spring, autumn and winter
[A15] We will modify the part to "*We describe the spatial variation of surface snow SSA measured during the four traverses between S16 and Dome Fuji in the austral summer and ...*"

323: meteorological events are by definition short-term. Delete "short-term".
[A16] We will remove the term "short-term" from "short-term meteorological events" throughout the manuscript.

431: I think it's interesting that glazed surfaces do not always lead to a very high variability?

[A17] If glazed surfaces predominate at a site, the SSA variability of 10 surfaces is not so high, which, I think, is what you refer to. The phrase might be confusing, and we will modify L418–420 in the original manuscript to "*The appearance of various surface morphologies including glazed surfaces results in higher SSA variability compared to the coastal and inland plateau regions (Table 1)*".

453: the magnitude and frequency of temperature gradients are much more important than absolute temperature

[A18] The importance of the magnitude and frequency of temperature gradients in SSA will be more emphasized in Sects. 4.1 and 4.5 (refer to [A07] of this file)

599: "under high pressure." there could also be clear days without high atmospheric pressure.

[A19] We will remove "under high pressure".

References:

Aoki, T., Kuchiki, K., Niwano, M., Kodama, Y., Hosaka, M., and Tanaka, T.: Physically based snow albedo model for calculating broadband albedos and the solar heating profile in snowpack for general circulation models, J. Geophys. Res. Atmos., 116, D11114, https://doi.org/10.1029/2010JD015507, 2011.

Azuma, N., Kameda, T., Nakayama, Y., Tanaka, Y., Yoshimi, H., Furukawa, T., and Ageta, Y.: Glaciological data collected by the 36th Japanese Antarctic Research Expedition during 1995-1996, JARE data reports. Glaciology, 26, 1–83, https://doi.org/10.15094/00004956, 1997.

Grenfell, T. C., Warren, S. G., and Mullen, P. C.: Reflection of solar radiation by the Antarctic snow surface at ultraviolet, visible, and near-infrared wavelengths, J. Geophys. Res. Atmos., 99, 18669–18684, https://doi.org/10.1029/94JD01484, 1994.

Hori, M., Murakami, H., Miyazaki, R., Honda, Y., Nasahara, K., Kajiwara, K., Nakajima, T. Y., Irie, H., Toratani, M., Hirawake, T., and Aoki, T.: GCOM-C Data Validation Plan for Land, Atmosphere, Ocean, and Cryosphere, Trans. JSASS Aerospace Tech. Japan, 16, 218–223, https://doi.org/10.2322/tastj.16.218, 2018.

Ishimoto, H., Adachi, S., Yamaguchi, S., Tanikawa, T., Aoki, T., and Masuda, K.: Snow particles extracted from X-ray computed microtomography imagery and their single-scattering properties, J. Quant. Spectrosc. Radiat. Transfer, 209, 113–128, https://doi.org/10.1016/j.jqsrt.2018.01.021, 2018.

Kinase, T., Adachi, K., Oshima, N., Goto-Azuma, K., Ogawa-Tsukagawa, Y., Kondo, Y., Moteki, N., Ohata, S., Mori, T., Hayashi, M., Hara, K., Kawashima, H., and Kita, K.: Concentrations and Size Distributions

of Black Carbon in the Surface Snow of Eastern Antarctica in 2011, J. Geophys. Res. Atmos., 125, e2019JD030737, https://doi.org/10.1029/2019JD030737, 2020.

Kokhanovsky, A., Rozanov, V. V., Aoki, T., Odermatt, D., Brockmann, C., Krüger, O., Bouvet, M., Drusch, M., and Hori, M.: Sizing snow grains using backscattered solar light, Int. J. Remote Sens., 32, 6975–7008, https://doi.org/10.1080/01431161.2011.560621, 2011.

Kokhanovsky, A., Lamare, M., Danne, O., Brockmann, C., Dumont, M., Picard, G., Arnaud, L., Favier, V., Jourdain, B., Le Meur, E., Di Mauro, B., Aoki, T., Niwano, M., Rozanov, V., Korkin, S., Kipfstuhl, S., Freitag, J., Hoerhold, M., Zuhr, A., Vladimirova, D., Faber, A.-K., Steen-Larsen, H. C., Wahl, S., Andersen, J. K., Vandecrux, B., van As, D., Mankoff, K. D., Kern, M., Zege, E., and Box, J. E.: Retrieval of Snow Properties from the Sentinel-3 Ocean and Land Colour Instrument, Remote Sens., 11, 2280, https://doi.org/10.3390/rs11192280, 2019.

Marbouty, D.: An Experimental Study of Temperature-Gradient Metamorphism, J. Glaciol., 26, 303–312, https://doi.org/10.3189/S0022143000010844, 1980.

Robledano, A., Picard, G., Dumont, M., Flin, F., Arnaud, L., and Libois, Q.: Unraveling the optical shape of snow, Nat Commun, 14, 3955, https://doi.org/10.1038/s41467-023-39671-3, 2023.

Warren, S. G. and Wiscombe, W. J.: A Model for the Spectral Albedo of Snow. II: Snow Containing Atmospheric Aerosols, J. Atmos. Sci., 37, 2734–2745, https://doi.org/10.1175/1520-0469(1980)037<2734:AMFTSA>2.0.CO;2, 1980.

Warren, S. G., Brandt, R. E., and Grenfell, T. C.: Visible and near-ultraviolet absorption spectrum of ice from transmission of solar radiation into snow, Appl. Opt., AO, 45, 5320–5334, https://doi.org/10.1364/AO.45.005320, 2006.

Zhou, X., Li, S., and Stamnes, K.: Effects of vertical inhomogeneity on snow spectral albedo and its implication for optical remote sensing of snow, J. Geophys. Res. Atmos., 108, https://doi.org/10.1029/2003JD003859, 2003.

---

## Author Comment (AC2)

The authors would like to thank the referee for acknowledging the value of our study and providing valuable feedback on our manuscript. We addressed all of your comments. Our responses are provided in blue, and the revised parts of the manuscript are marked in red.

Reviewer #2: Anonymous Referee

5    The study describes a rich dataset of measurements of the specific surface area (SSA) of snow acquired during four traverses from the coast to the inland plateau of Antarctica and during a 13-day stay at Dome Fuji Station, on the plateau. The authors present the measurement locations, methods and auxiliary data in a reproducible way. The drivers of the variations of the SSA observed are then discussed. The article is clear, the results are well presented, and figures are effective. Overall, it nicely highlights the complexity

10   of the surface processes impacting the snow optical and microstructural properties over the Antarctic ice sheet during summer.

However, some details about the measurement procedure should be cleared in the text. At line 193 the SSA is said to be measured "at 10 different surfaces, spaced 2 m apart along a transect perpendicular to the predominant wind direction". First, a more detailed scheme of the measurement area could help

15   improve both understanding of the text and potential for satellite product validation.

Because of limited observation time during traverses, we simply set the 10 measurement surfaces by taking four steps forward between the surfaces, and the flat area in front of the toes was measured. I trained in advance to keep my stride length to a total of 2 m. While this scheme may not ensure accurate 2 m intervals, we believe that 10 surfaces were sufficiently randomly selected, not affecting our

20   conclusions.

      We will modify L193–194 in the original manuscript to "*At each observation site, we measured surface snow SSA at 10 different surfaces spaced roughly 2 m apart along a transect, by taking four steps forward and measuring the surface in front of the toes.*"

Second, why the choice of the transect perpendicular to the predominant wind direction? Is it related to

25   the presence of wind-induced bedforms?

Yes. The predominant wind direction was determined by surface morphologies such as dunes and sastrugi, which typically extend parallel to the wind direction along our traverse route. To avoid measurement surfaces from being biased toward a specific bedform extending parallel to the wind direction, we positioned the transect perpendicular to it.

30       We will add "*The transect was positioned perpendicular to the predominant wind direction or the direction in which dunes and sastrugi extended, to prevent measurement surfaces from being biased toward a bedform.*" to L194 in the original text.

Third, how was the sampling carried out over rough surfaces (surfaces c. and d. in section 2.2.2), considering that erosion bedforms present different snow properties according to their exposition to the wind (Sommer et al. 2018)?

At such rough surfaces, five SSA measurements were performed within a flat area of $\sim 0.05$ m$^2$ (e.g., the side of sastrugi), which appears to have similar characteristics (e.g., density, grain size, and elapsed time after deposition). While a deposition and erosion surface may present SSA variation over several meter scales, as noted by Sommer et al. (2018), the variability of our five measurements within $\sim 0.05$ m$^2$ is small (3.5%). Investigating SSA variability within bedforms at several meter scales would be future work.

We will modify L194–195 in the original text to "*For each of the 10 surfaces, we conducted five measurements within a flat area of approximately 0.05 m$^2$ that appears to have similar snow properties, by shifting the measurement positions by approximately 0.1 m, and calculated their mean value.*"

Finally, it would be interesting to have more details about the measurement uncertainty and how it compares to the SSA variability observed along the transects.

The measurement uncertainty will be elaborated in L201–203; "*The relative SD of the five measurements at a surface, which represents the random error in a SSA measurement, is 3.5 ± 2.5 % (the average and SD for $\sim 2150$ surfaces). The absolute error in the HISSGraS measurements has been evaluated as 23.0% (Aoki et al., 2023). This value represents the relative root mean square error of HISSGraS data for 30 snow samples with SSA of 5–30 m$^2$ kg$^{-1}$ collected in Hokkaido, Japan, compared to the reference SSA from the CH4 adsorption method (accuracy of 12%) (Legagneux et al., 2002)*".

The latter error (23.0%) of HISSGraS data includes a systematic error from SSA from the CH$_4$ absorption method and other error sources (e.g., the time difference between the two SSA measurements) (Aoki et al., 2023). In the relative comparison among measured SSA in this study, a random error of 3.5% (e.g., 0.7 m$^2$ kg$^{-1}$ for 20 m$^2$ kg$^{-1}$) should be considered. This error is smaller than the SD for 10 surfaces along the transects (typically 1–5 m$^2$ kg$^{-1}$ for a 10-surface mean of $\sim 20$ m$^2$ kg$^{-1}$), suggesting that the SSA variability observed along the transects is statistically significant. We will describe this in L317–319 in the original text; "*The appearance of various surface morphologies between 380–680 km, including fresh and aged deposition surfaces and glazed surface, results in the significant variability of SSA in the range. This variability is evident even within a 20 m transect at each observation site, with a SD of 10 surfaces (up to 20 m$^2$ kg$^{-1}$) exceeding the random error at a surface (3.5%).*".

Some minor comments:

Line 56 : clarify the difference between sublimation that decreases the snow SSA and that increases the snow SSA (line 58).

We will elaborate these two cases; "*It may decrease snow SSA by sublimating fine needles or branches of dendric crystals in freshly fallen snow with a high SSA, transforming them into rounded grains or*

*causing them to disappear (Cabanes et al., 2002), and by eroding deposited snow, thereby exposing aged*
*snow with a lower SSA (e.g., Lenaerts et al., 2017). Conversely, the wind may increase the SSA through*
*the sublimation of snow grains into smaller particles and fragmentation of drifting snow crystals,*
*creating new surfaces (Domine et al., 2009)."*

Line 60 : a brief description of snow bedforms that result from wind-induced redistribution could be
introduced here, with the heterogeneous distribution.

We will modify the sentence to "*Strong winds further contribute to snow redistribution and*
*heterogeneous deposition (Kameda et al., 2008; Picard et al., 2019), forming dunes or snowdrifts that*
*elongate either parallel to or across the wind direction over several meters while exposing old snow at*
*the surface in eroded areas (Filhol and Sturm, 2015; Sommer et al., 2018).*"

Figure 3 : a horizontal grid could improve the clarity of the figure, as most variations of the SSA in time
are small

Thank you for the suggestion. We will add a horizontal grid to Figure 3.

Line 313 : the definition of erosion surfaces as "aged deposition surfaces" is confusing with respect to the
definitions in section 2.2.2

Sorry for the confusion in our manuscript. We will revise it to "*Erosion surfaces *
* are predominantly observed in the coastal region , aged deposition*
*surfaces are observed in the inland plateau region, while ...*". Here, "*aged deposition surfaces*" in
parenthesis did not define "*Erosion surfaces*", but was the subject of "*(inland plateau region)*".

Line 466 : the sentence "Also, the observed SSA     … air temperatures." is not clear, please rephrase

We will rephrase the sentence; "*SSA correlates non-linearly with air temperature, with large spatial*
*variation in SSA observed in the lower temperature range (~ −35 to −15°C) and less variation observed*
*in the higher temperature range (~ −15 to 0°C)."*